# $\beta$-INTACT-VAE: IDENTIFYING AND ESTIMATING CAUSAL EFFECTS UNDER LIMITED OVERLAP

**Pengzhou (Abel) Wu & Kenji Fukumizu**
Department of Statistical Science, The Graduate University for Advanced Studies
& The Institute of Statistical Mathematics
Tachikawa, Tokyo
{wu.pengzhou,fukumizu}@ism.ac.jp

## ABSTRACT

As an important problem in causal inference, we discuss the identification and estimation of treatment effects (TEs) under limited overlap; that is, when subjects with certain features belong to a single treatment group. We use a latent variable to model a prognostic score which is widely used in biostatistics and sufficient for TEs; i.e., we build a generative prognostic model. We prove that the latent variable recovers a prognostic score, and the model identifies individualized treatment effects. The model is then learned as $\beta$-Intact-VAE—a new type of variational autoencoder (VAE). We derive the TE error bounds that enable representations balanced for treatment groups conditioned on individualized features. The proposed method is compared with recent methods using (semi-)synthetic datasets.

## 1 INTRODUCTION

Causal inference (Imbens & Rubin, 2015; Pearl, 2009), i.e, inferring causal effects of interventions, is a fundamental field of research. In this work, we focus on treatment effects (TEs) based on a set of observations comprising binary labels $T$ for treatment/control (non-treated), outcome $Y$, and other covariates $X$. Typical examples include estimating the effects of public policies or new drugs based on the personal records of the subjects. The fundamental difficulty of causal inference is that we never observe *counterfactual* outcomes that would have been if we had made the other decision (treatment or control). While randomized controlled trials (RCTs) control biases through randomization and are ideal protocols for causal inference, they often have ethical and practical issues, or suffer from expensive costs. Thus, causal inference from observational data is important.

Causal inference from observational data has other challenges as well. One is *confounding*: there may be variables, called confounders, that causally affect both the treatment and the outcome, and spurious correlation/bias follows. The other is the systematic *imbalance* (difference) of the distributions of the covariates between the treatment and control groups—that is, $X$ depends on $T$, which introduces bias in estimation. A majority of studies on causal inference, including the current work, have relied on unconfoundedness; this means that the confounding can be controlled by conditioning on the covariates. The more covariates are collected the more likely unconfoundedness holds; however, more covariates tends to introduce a stronger imbalance between treatment and control.

The current work studies the issue of imbalance in estimating individualized TEs conditioned on $X$. Classical approaches aim for *covariate balance*, $X$ independent of $T$, by matching and re-weighting (Stuart, 2010; Rosenbaum, 2020). Machine learning methods have also been exploited; there are semi-parametric methods—e.g., Van der Laan & Rose (2018, TMLE)—which improve finite sample performance, as well as non-parametric methods—e.g., Wager & Athey (2018, CF). Notably, from Johansson et al. (2016), there has been a recent increase in interest in *balanced representation learning* (BRL) to learn representations $Z$ of the covariates, such that $Z$ independent of $T$.

The most serious form of imbalance is the *limited (or weak) overlap of covariates*, which means that sample points with certain covariate values belong to a single treatment group. In this case, a straightforward estimation of TEs is not possible at non-overlapping covariate values due to lack of data. There are works that provide robustness to limited overlap (Armstrong & Kolesár, 2021), trim non-overlapping data points (Yang & Ding, 2018), weight data points by overlap (Li & Li, 2019), or study convergence rates depending on overlap (Hong et al., 2020). Limited overlap is particularly relevant to machine learning methods that exploit high-dimensional covariates. This is because, with higher-dimensional covariates, overlap is harder to satisfy and verify (D'Amour et al., 2020).

To address imbalance and limited overlap, we use a prognostic score (Hansen, 2008); it is a sufficient statistic of outcome predictors and is among the key concepts of sufficient scores for TE estimation. As a function of covariates, it can map some non-overlapping values to an overlapping value in a space of lower-dimensions. For individualized TEs, we consider *conditionally balanced representation $Z$*, such that $Z$ is independent of $T$ given $X$—which, as we will see, is a necessary condition for a balanced prognostic score. Moreover, prognostic score modeling can benefit from methods in predictive analytics and exploit rich literature, particularly in medicine and health (Hajage et al., 2017). Thus, it is promising to combine the predictive power of prognostic modeling and machine learning. With this idea, our method builds on a generative prognostic model that models the prognostic score as a latent variable and factorizes to the score distribution and outcome distribution.

As we consider latent variables and causal inference, *identification* is an issue that must be discussed before estimation is considered. "Identification" means that the parameters of interest (in our case, representation function and TEs) are uniquely determined and expressed using the true observational distribution. Without identification, a consistent estimator is impossible to obtain, and a model would fail silently; in other words, the model may fit perfectly but will return an estimator that converges to a wrong one, or does not converge at all (Lewbel, 2019, particularly Sec. 8). Identification is even more important for causal inference; because, unlike usual (non-causal) model misspecification, causal assumptions are often unverifiable through observables (White & Chalak, 2013). Thus, it is critical to specify the theoretical conditions for identification, and then the applicability of the methods can be judged by knowledge of an application domain.

A major strength of our generative model is that the latent variable is identifiable. This is because the factorization of our model is naturally realized as a combination of identifiable VAE (Khemakhem et al., 2020a, iVAE) and conditional VAE (Sohn et al., 2015, CVAE). Based on model identifiability, we develop two identification results for individualized TEs under limited overlap. A similar VAE architecture was proposed in Wu & Fukumizu (2020b); the current study is different in setting, theory, learning objective, and experiments. The previous work studies unobserved confounding but not limited overlap, with different set of assumptions and identification theories. The current study further provides bounds on individualized TE error, and the bounds justify a conditionally balancing term controlled by hyperparameter $\beta$, as an interpolation between the two identifications.

In summary, we study the identification (Sec. 3) and estimation (Sec. 4) of individualized TEs under limited overlap. Our approach is based on recovering prognostic scores from observed variables. To this end, our method exploits recent advances in identifiable representation—particularly iVAE. The code is in Supplementary Material, and the proofs are in Sec. A. Our main contributions are:
1) TE identification under limited overlap of $X$, via prognostic scores and an identifiable model;
2) bounds on individualized TE error, which justify our conditional BRL;
3) a new regularized VAE, $\beta$-Intact-VAE, realizing the identification and conditional balance;
4) experimental comparison to the state-of-the-art methods on (semi-)synthetic datasets.

## 1.1 RELATED WORK

**Limited overlap.** Under limited overlap, Luo et al. (2017) estimate the average TE (ATE) by reducing covariates to a linear prognostic score. Farrell (2015) estimates a constant TE under a partial linear outcome model. D'Amour & Franks (2021) study the identification of ATE by a general class of scores, given the (linear) propensity score and prognostic score. Machine learning studies on this topic have focused on finding overlapping regions (Oberst et al., 2020; Dai & Stultz, 2020), or indicating possible failure under limited overlap (Jesson et al., 2020), but not remedies. An exception is Johansson et al. (2020), which provides bounds under limited overlap. To the best of our knowledge, our method is the first machine learning method that provides identification under limited overlap.

**Prognostic scores** have been recently combined with machine learning approaches, mainly in the biostatistics community. For example, Huang & Chan (2017) estimate individualized TE by reducing covariates to a linear score which is a joint propensity-prognostic score. Tarr & Imai (2021) use SVM to minimize the worst-case bias due to prognostic score imbalance. However, in the machine learning community, few methods consider prognostic scores; Zhang et al. (2020a) and Hassanpour & Greiner (2019) learn outcome predictors, without mentioning prognostic score—while Johansson et al. (2020) conceptually, but not formally, connects BRL to prognostic score. Our work is the first to formally connect generative learning and prognostic scores for TE estimation.

**Identifiable representation.** Recently, independent component analysis (ICA) and representation learning—both ill-posed inverse problems—meet together to yield nonlinear ICA and identifiable representation; for example, using VAEs (Khemakhem et al., 2020a), and energy models (Khemakhem et al., 2020b). The results are exploited in causal discovery (Wu & Fukumizu, 2020a) and out-of-distribution (OOD) generalization (Sun et al., 2020). This study is the first to explore identifiable representations in TE identification.

**BRL and related methods** amount to a major direction. Early BRL methods include BLR/BNN (Johansson et al., 2016) and TARnet/CFR (Shalit et al., 2017). In addition, Yao et al. (2018) exploit the local similarity between data points. Shi et al. (2019) use similar architecture to TARnet, considering the importance of treatment probability. There are also methods that use GAN (Yoon et al., 2018, GANITE) and Gaussian processes (Alaa & van der Schaar, 2017). Our method shares the idea of BRL, and further extends to conditional balance—which is natural for individualized TE.

**More.** Our work lays conceptual and theoretical foundations of VAE methods for TEs (e.g., CEVAE Louizos et al., 2017; Lu et al., 2020). See Sec. D for more related works, there we also make detailed comparisons to CFR and CEVAE, which are well-known machine learning methods.

## 2    SETUP AND PRELIMINARIES

### 2.1    COUNTERFACTUALS, TREATMENT EFFECTS, AND IDENTIFICATION

Following Imbens & Rubin (2015), we assume there exist *potential outcomes* $Y(t) \in \mathbb{R}^d, t \in \{0, 1\}$. $Y(t)$ is the outcome that would have been observed if the treatment value $T = t$ was applied. We see $Y(t)$ as the hidden variables that give the *factual outcome* $Y$ under *factual assignment* $T = t$. Formally, $Y(t)$ is defined by the *consistency of counterfactuals*: $Y = Y(t)$ if $T = t$; or simply $Y = Y(T)$. The *fundamental problem of causal inference* is that, for a unit under research, we can observe only one of $Y(0)$ or $Y(1)$—w.r.t. the treatment value applied. That is, "factual" refers to $Y$ or $T$, which is *observable*; or estimators built on the observables. We also observe relevant covariate(s) $X \in \mathcal{X} \subseteq \mathbb{R}^m$, which is associated with individuals, with distribution $\mathcal{D} := (X, Y, T) \sim p(\boldsymbol{x}, \boldsymbol{y}, t)$. We use upper-case (e.g. $T$) to denote random variables, and lower-case (e.g. $t$) for realizations.

The expected potential outcome is denoted by $\mu_t(\boldsymbol{x}) = \mathbb{E}(Y(t)|X = \boldsymbol{x})$ conditioned on $X = \boldsymbol{x}$. The estimands in this work are the conditional ATE (CATE) and ATE, defined, respectively, by:

$$\tau(\boldsymbol{x}) = \mu_1(\boldsymbol{x}) - \mu_0(\boldsymbol{x}), \quad \nu = \mathbb{E}(\tau(X)). \tag{1}$$

CATE is seen as an *individual-level*, personalized, treatment effect, given highly discriminative $X$.

Standard results (Rubin, 2005)(Hernan & Robins, 2020, Ch. 3) show sufficient conditions for TE identification in general settings. They are *Exchangeability*: $Y(t) \perp\!\!\!\perp T|X$, and *Overlap*: $p(t|\boldsymbol{x}) > 0$ for any $\boldsymbol{x} \in \mathcal{X}$. Both are required for $t \in \{0, 1\}$. When $t$ appears in statements without quantification, we always mean "for both $t$". Often, *Consistency* is also listed; however, as mentioned, it is better known as the well-definedness of counterfactuals. Exchangeability means, just as in RCTs, but additionally given $X$, that there is no correlation between factual $T$ and potential $Y(t)$. Note that the popular assumption $Y(0), Y(1) \perp\!\!\!\perp T|X$ is stronger than $Y(t) \perp\!\!\!\perp T|X$ and is not necessary for identification (Hernan & Robins, 2020, pp. 15). Overlap means that the supports of $p(\boldsymbol{x}|t = 0)$ and $p(\boldsymbol{x}|t = 1)$ should be the same, and this ensures that there are data for $\mu_t(\boldsymbol{x})$ on any $(\boldsymbol{x}, t)$.

We rely on consistency and exchangeability, but in Sec. 3.2, will relax the condition of the overlapping covariate to allow some non-overlapping values $\boldsymbol{x}$—that is, covariate $X$ is *limited-overlapping*. In this paper, we also discuss overlapping variables other than $X$ (e.g., prognostic scores), and provide a definition for any random variable $V$ with support $\mathcal{V}$ as follows:

**Definition 1.** $V$ is *Overlapping* if $p(t|V = \boldsymbol{v}) > 0$ for any $t \in \{0, 1\}, \boldsymbol{v} \in \mathcal{V}$. If the condition is violated at some value $\boldsymbol{v}$, then $\boldsymbol{v}$ is *non-overlapping* and $V$ is *limited-overlapping*.

### 2.2    PROGNOSTIC SCORES

Our method aims to recover a prognostic score (Hansen, 2008), adapted to account for both $t$ as in Definition 2. On the other hand, balancing scores (Rosenbaum & Rubin, 1983) $\boldsymbol{b}(X)$ are defined by $T \perp\!\!\!\perp X|\boldsymbol{b}(X)$, of which the propensity score $p(t = 1|X)$ is a special case. See Sec. B.1 for detail.

**Definition 2.** A *PGS* is $\{\mathfrak{p}(X,t)\}_{t\in\{0,1\}}$ such that $Y(t)\perp\!\!\!\perp X|\mathfrak{p}(X,t)$, where $\mathfrak{p}(\boldsymbol{x},t)$ ($\mathfrak{p}_t(\boldsymbol{x})$ hereafter) is a function defined on $\mathcal{X}\times\{0,1\}$. A PGS is called *balanced* (and a *bPGS*) if $\mathfrak{p}_0 = \mathfrak{p}_1$.

We say a PGS is overlapping, if *both* $\mathfrak{p}_0(X)$ and $\mathfrak{p}_1(X)$ are overlapping. Obviously, a bPGS $\mathfrak{p}(X)$ is a conditionally balanced representation (defined as $Z\perp\!\!\!\perp T|X$ in Introduction) and is thus named. We often write $t$ of the function argument in subscripts.

We use bPGS or PGS to construct representations for CATE estimation. **Why not balancing scores?** While balancing scores $\boldsymbol{b}(X)$ have been widely used in causal inference, PGSs are more suitable for discussing overlap. Our purpose is to recover an overlapping score for limited-overlapping $X$. It is known that overlapping $\boldsymbol{b}(X)$ implies overlapping $X$ (D'Amour et al., 2020), which counters our purpose. In contrast, overlapping bPGS does not imply overlapping $\boldsymbol{b}(X)$. **Example.** Let $T = \mathbb{I}(X+\boldsymbol{\epsilon}>0)$ and $Y = \boldsymbol{f}(|X|,T)+\mathbf{e}$, where $\mathbb{I}$ is the indicator function, $\boldsymbol{\epsilon}$ and $\mathbf{e}$ are exogenous zero-mean noises, and the support of $X$ is on the entire real line while $\boldsymbol{\epsilon}$ is bounded. Now, $X$ itself is a balancing score and $|X|$ is a bPGS; and $|X|$ is overlapping but $X$ is not. Moreover, with theoretical and experimental evidence, it is recently conjectured that PGSs maximize overlap among a class of sufficient scores, including $\boldsymbol{b}(X)$ (D'Amour & Franks, 2021). In general, Hajage et al. (2017) show that prognostic score methods perform better—or as well as—propensity score methods.

Below is a corollary of Proposition 5 in Hansen (2008); note that $\mathfrak{p}_t(X)$ satisfies exchangeability.

**Proposition 1** (Identification via PGS). *If $\mathfrak{p}_t(X)$ is a PGS and $Y|\mathfrak{p}_{\hat{t}}(X), T \sim p_{Y|\mathfrak{p}_{\hat{t}},T}(\boldsymbol{y}|P,t)$ where $\hat{t}\in\{0,1\}$ is a counterfactual assignment, then CATE and ATE are identified, using* (1) *and*

$$\mu_{\hat{t}}(\boldsymbol{x}) = \mathbb{E}(Y(\hat{t})|\mathfrak{p}_{\hat{t}}(X), X=\boldsymbol{x}) = \mathbb{E}(Y|\mathfrak{p}_{\hat{t}}(\boldsymbol{x}), T=\hat{t}) = \int p_{Y|\mathfrak{p}_{\hat{t}},T}(\boldsymbol{y}|\mathfrak{p}_{\hat{t}}(\boldsymbol{x}),\hat{t})\boldsymbol{y}d\boldsymbol{y} \quad (2)$$

With the knowledge of $\mathfrak{p}_t$ and $p_{Y|\mathfrak{p}_{\hat{t}},T}$, we choose one of $\mathfrak{p}_0, \mathfrak{p}_1$ and set $t = \hat{t}$ in the density function, w.r.t the $\mu_{\hat{t}}$ of interest. This counterfactual assignment resolves the problem of non-overlap at $\boldsymbol{x}$. Note that a sample point with $X=\boldsymbol{x}$ may not have $T=\hat{t}$.

We consider additive noise models for $Y(t)$, which ensures the existence of PGSs.

**(G1)**[1] (Additive noise model) the data generating process (DGP) for $Y$ is $Y = \boldsymbol{f}^*(\mathfrak{m}(X,T),T) + \mathbf{e}$ where $\boldsymbol{f}^*, \mathfrak{m}$ are functions and $\mathbf{e}$ is a zero-mean exogenous (external) noise.

The DGP is causal and defines potential outcomes by $Y(t) := \boldsymbol{f}_t^*(\mathfrak{m}_t(X)) + \mathbf{e}$, and specifies $\mathfrak{m}(X,T)$, $T$, and $\mathbf{e}$ as the only direct causes of $Y$. Particularly, $\mathfrak{m}_t(X)$ is a sufficient statistics of $X$ for $Y(t)$. For example, 1) $\mathfrak{m}_t(X)$ can be the component(s) of $X$ that affect $Y(t)$ directly, or 2) if $Y(t)|X$ follows a generalized linear model, then $\mathfrak{m}_t(X)$ can be the linear predictor of $Y(t)$.

Under **(G1)**, 1) $\mathfrak{m}_t(X)$ is a PGS; 2) $\mu_t(X) = \boldsymbol{f}_t^*(\mathfrak{m}_t(X))$ is a PGS; 3) $X$ is a (trivial) bPGS; and 4) $\mathfrak{u}(X) := (\mu_0(X),\mu_1(X))$ is a bPGS. The **essence of our method** is to recover the PGS $\mathfrak{m}_t(X)$ as a representation, assuming $\mathfrak{m}_t(X)$ is not higher-dimensional than $Y$ and approximately balanced. Note that $\mu_t(X)$, our final target, is a low-dimensional PGS but not balanced, and we estimate it conditioning on the approximate bPGS $\mathfrak{m}_t(X)$.

## 3 IDENTIFICATION UNDER GENERATIVE PROGNOSTIC MODEL

In Sec. 3.1, we specify the generative prognostic model $p(\boldsymbol{y},\boldsymbol{z}|\boldsymbol{x},t)$, and show its identifiability. In Sec. 3.2, we prove the identification of CATEs, which is one of our main contributions. The theoretical analysis involves only our generative model (i.e., prior and decoder), but not the encoder. The encoder is not part of the generative model and is involved as an approximate posterior in the estimation, which is studied in Sec. 4.

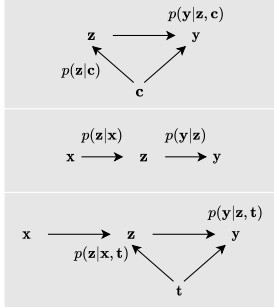

### 3.1 MODEL, ARCHITECTURE, AND IDENTIFIABILITY

Our goal is to build a model that can be learned by VAE from observational data to obtain a PGS, or better, a bPGS, via the latent variable $Z$. The generative prognostic model of the proposed method is in (3),

Figure 1: CVAE, iVAE, and Intact-VAE: Graphical models of the decoders.

---

[1]The labels **G**, **M**, or **D** mean Generating process (of $Y$), probabilistic Model, or Distribution (of $X$). We introduce assumptions when appropriate but compile them in one place in Sec. C.1.

where $\boldsymbol{\theta} := (\boldsymbol{f}, \boldsymbol{h}, \boldsymbol{k})$ contains the functional parameters. The first factor $p_{\boldsymbol{f}}(\boldsymbol{y}|\boldsymbol{z}, t)$, our decoder, models $p_{Y|\mathfrak{p}_t, T}(\boldsymbol{y}|P, t)$ in (2) and is an additive noise model, with $\boldsymbol{\epsilon} \sim p_{\boldsymbol{\epsilon}}$ as the exogenous noise. The second factor $p_{\boldsymbol{\lambda}}(\boldsymbol{z}|\boldsymbol{x}, t)$, our conditional prior, models $\mathfrak{p}_T(X)$ and is a factorized Gaussian, with $\boldsymbol{\lambda}_T(X) := \text{diag}^{-1}(\boldsymbol{k}_T(X))(\boldsymbol{h}_T(X), -\frac{1}{2})^T$ as its natural parameter in the exponential family, where $\text{diag}()$ gives a diagonal matrix from a vector.

$$p_{\boldsymbol{\theta}}(\boldsymbol{y}, \boldsymbol{z}|\boldsymbol{x}, t) = p_{\boldsymbol{f}}(\boldsymbol{y}|\boldsymbol{z}, t)p_{\boldsymbol{\lambda}}(\boldsymbol{z}|\boldsymbol{x}, t),$$
$$p_{\boldsymbol{f}}(\boldsymbol{y}|\boldsymbol{z}, t) = p_{\boldsymbol{\epsilon}}(\boldsymbol{y} - \boldsymbol{f}_t(\boldsymbol{z})), \quad p_{\boldsymbol{\lambda}}(\boldsymbol{z}|\boldsymbol{x}, t) \sim \mathcal{N}(\boldsymbol{z}; \boldsymbol{h}_t(\boldsymbol{x}), \text{diag}(\boldsymbol{k}_t(\boldsymbol{x}))). \tag{3}$$

We denote $n := \dim(Z)$. For inference, the ELBO is given by the standard variational lower bound

$$\log p(\boldsymbol{y}|\boldsymbol{x}, t) \geq \mathbb{E}_{\boldsymbol{z} \sim q} \log p_{\boldsymbol{f}}(\boldsymbol{y}|\boldsymbol{z}, t) - D_{\text{KL}}(q(\boldsymbol{z}|\boldsymbol{x}, \boldsymbol{y}, t) \| p_{\boldsymbol{\lambda}}(\boldsymbol{z}|\boldsymbol{x}, t)). \tag{4}$$

Note that the encoder $q$ conditions on all the observables $(X, Y, T)$; this fact plays an important role in Sec. 4.1. Full parameterization of the encoder and decoder is also given in Sec. 4.1. This architecture is called *Intact-VAE* (*Id*entifiable *t*reatment-*c*ondi*t*ional VAE). See Figure 1 for comparison in terms of graphical models (which have *not* causal implications here). See Sec. C.2 for more expositions and Sec. B.2 for basics of VAEs.

Our model identifiability extends the theory of iVAE, and the following conditions are inherited.

**(M1)** i) $\boldsymbol{f}_t$ is injective, and ii) $\boldsymbol{f}_t$ is differentiable.

**(D1)** $\boldsymbol{\lambda}_t(X)$ is non-degenerate, i.e., the linear hull of its support is $2n$-dimensional.

Under **(M1)** and **(D1)**, we obtain the following identifiability of the parameters in the model: if $p_{\boldsymbol{\theta}}(\boldsymbol{y}|\boldsymbol{x}, t) = p_{\boldsymbol{\theta}'}(\boldsymbol{y}|\boldsymbol{x}, t)$, we have, for any $\boldsymbol{y}_t$ in the image of $\boldsymbol{f}_t$:

$$\boldsymbol{f}_t^{-1}(\boldsymbol{y}_t) = \text{diag}(\boldsymbol{a})\boldsymbol{f}_t'^{-1}(\boldsymbol{y}_t) + \boldsymbol{b} =: \mathcal{A}_t(\boldsymbol{f}_t'^{-1}(\boldsymbol{y}_t)) \tag{5}$$

where $\text{diag}(\boldsymbol{a})$ is an invertible $n$-diagonal matrix and $\boldsymbol{b}$ is an $n$-vector, both of which depend on $\boldsymbol{\lambda}_t(\boldsymbol{x})$ and $\boldsymbol{\lambda}_t'(\boldsymbol{x})$. The essence of the result is that $\boldsymbol{f}_t' = \boldsymbol{f}_t \circ \mathcal{A}_t$; that is, $\boldsymbol{f}_t$ can be identified (learned) up to an affine transformation $\mathcal{A}_t$. See Sec. A for the proof and a relaxation of **(D1)**. In this paper, symbol $'$ (prime) always indicates another parameter (variable, etc.): $\boldsymbol{\theta}' = (\boldsymbol{f}', \boldsymbol{\lambda}')$.

## 3.2 IDENTIFICATIONS UNDER LIMITED-OVERLAPPING COVARIATE

In this subsection, we present two results of CATE identification based on the recovery of equivalent bPGS and PGS, respectively. Since PGSs are functions of $X$, the theory assumes a noiseless prior for simplicity, i.e., $\boldsymbol{k}(X) = \boldsymbol{0}$; the prior $Z_{\boldsymbol{\lambda}, t} \sim p_{\boldsymbol{\lambda}}(\boldsymbol{z}|\boldsymbol{x}, t)$ degenerates to function $\boldsymbol{h}_t(X)$.

PGSs with dimensionality lower than or equal to $d = \dim(Y)$ are essential to address limited overlapping, as shown below. We set $n = d$ because $\mu_t$ is a PGS of the same dimension as $Y$ under **(G1)**. In practice, $n = d$ means that we seek a low-dimensional representation of $X$. We introduce

**(G1')** (Low-dimensional PGS) **(G1)** is true, and $\mu_t = \boldsymbol{j}_t \circ \mathfrak{p}_t$ for some $\mathfrak{p}_t$ and injective $\boldsymbol{j}_t$,

which is equivalent to **(G1)** because $\mu_t = \boldsymbol{j}_t \circ \mathfrak{p}_t$ is trivially satisfied with $\boldsymbol{j}_t$ is identity and $\mathfrak{p}_t = \mu_t$. **(G1')** is used instead in this subsection. First, it explicitly restricts $\dim(\mathfrak{p}_t)$ via injectivity, which ensures that $n = \dim(Y) \geq \dim(\mathfrak{p}_t)$. Second, it reminds us that, possibly, the decomposition is not unique; and, clearly, all $\mathfrak{p}_t$ that satisfy **(G1')** are PGSs. For example, if $\boldsymbol{f}_t^*$ is injective, then $\boldsymbol{j}_t = \boldsymbol{f}_t^*$ and $\mathfrak{p}_t = \mathfrak{m}_t$ satisfies $\mu_t = \boldsymbol{j}_t \circ \mathfrak{p}_t$. Finally, it is then natural to introduce

**(G2)** (Low-dimensional bPGS) **(G1)** is true, and $\mu_t = \boldsymbol{j}_t \circ \mathfrak{p}$ for some $\mathfrak{p}$ and injective $\boldsymbol{j}_t$,

which is stronger than **(G1)**, gives bPGS $\mathfrak{p}(X)$, and ensures that $n \geq \dim(\mathfrak{p})$. **(G2)** is satisfied if $\boldsymbol{f}_t^*$ is injective and $\mathfrak{m}_0 = \mathfrak{m}_1$. **(G2)** implies $\mu_1 = \boldsymbol{i} \circ \mu_0$ where $\boldsymbol{i} := \boldsymbol{j}_1 \circ \boldsymbol{j}_0^{-1}$; in words, CATEs are given by $\mu_0$ and an invertible function. See Sec. C.3 for real-world examples and more discussions.

With **(G1')** or **(G2)**, overlapping $X$ can be relaxed to overlapping bPGS or PGS plus the following:

**(M2)** (Score partition preserving) For any $\boldsymbol{x}, \boldsymbol{x}' \in \mathcal{X}$, if $\mathfrak{p}_t(\boldsymbol{x}) = \mathfrak{p}_t(\boldsymbol{x}')$, then $\boldsymbol{h}_t(\boldsymbol{x}) = \boldsymbol{h}_t(\boldsymbol{x}')$.

Note that **(M2)** is only required for the optimal $\boldsymbol{h}$ specified in Proposition 2 or Theorem 1. The intuition is that $\mathfrak{p}_t$ maps each non-overlapping $\boldsymbol{x}$ to an overlapping value, and $\boldsymbol{h}_t$ preserves this property through learning. This is non-trivial because, for a given $t$, some values of $X$ are unobserved due to limited overlap. Thus, **(M2)** can be seen as a weak form of OOD generalization: the NNs for $\boldsymbol{h}$ can

learn the OOD score partition. While unnecessary for us, linear $\mathfrak{p}_t$ and $\boldsymbol{h}_t$ trivially imply **(M2)** and are often assumed, e.g., in Huang & Chan (2017); Luo et al. (2017); D'Amour & Franks (2021).

Our first identification, Proposition 2, relies on **(G2)** and our generative model, *without* model identifiability (so differentiable $\boldsymbol{f}_t$ is not needed).

**Proposition 2** (Identification via recovery of bPGS). *Suppose we have DGP **(G2)** and model* (3) *with $n = d$. Assume **(M1)-i)** and **(M3)** (PS matching) let $\boldsymbol{h}_0(X) = \boldsymbol{h}_1(X)$ and $\boldsymbol{k}(X) = \boldsymbol{0}$. Then, if $\mathbb{E}_{p_{\boldsymbol{\theta}}}(Y|X,T) = \mathbb{E}(Y|X,T)$, we have*

1) *(Recovery of bPGS) $\boldsymbol{z}_{\boldsymbol{\lambda},t} = \boldsymbol{h}_t(\boldsymbol{x}) = \boldsymbol{v}(\mathfrak{p}(\boldsymbol{x}))$ on overlapping $\boldsymbol{x}$,*
   *where $\boldsymbol{v} : \mathcal{P} \to \mathbb{R}^n$ is an injective function, and $\mathcal{P} \coloneqq \{\mathfrak{p}(\boldsymbol{x})|$overlapping $\boldsymbol{x}\}$;*
2) *(CATE identification) if $\mathfrak{p}(X)$ in **(G2)** is overlapping, and **(M2)** is satisfied, then*
   *$\mu_t(\boldsymbol{x}) = \hat{\mu}_t(\boldsymbol{x}) \coloneqq \mathbb{E}_{p_{\boldsymbol{\lambda}}(Z|\boldsymbol{x},t)}\mathbb{E}_{p_{\boldsymbol{f}}}(Y|Z,t) = \boldsymbol{f}_t(\boldsymbol{h}_t(\boldsymbol{x})), $ for any $t \in \{0,1\}$ and $\boldsymbol{x} \in \mathcal{X}$.*

In essence, i) the true DGP is identified up to an invertible mapping $\boldsymbol{v}$, such that $\boldsymbol{f}_t = \boldsymbol{j}_t \circ \boldsymbol{v}^{-1}$ and $\boldsymbol{h} = \boldsymbol{v} \circ \mathfrak{p}$; and ii) $\mathfrak{p}_t$ is recovered up to $\boldsymbol{v}$, and $Y(t) \perp\!\!\!\perp X | \mathfrak{p}_t(X)$ is preserved—with *same* $\boldsymbol{v}$ for both $t$. Theorem 1 below also achieves the essence i) and ii), under $\mathfrak{p}_0 \neq \mathfrak{p}_1$.

The existence of bPGS is preferred, because it satisfies overlap and **(M2)** more easily than PGS which requires the conditions for each of the two functions of PGS. However, the existence of low-dimensional bPGS is uncertain in practice when our knowledge of the DGP is limited. Thus, we depend on Theorem 1 based on the model identifiability to work under PGS which generally exists.

**Theorem 1** (Identification via recovery of PGS). *Suppose we have DGP **(G1')** and model* (3) *with $n = d$. For the model, assume **(M1)** and **(M3')** (Noise matching) let $p_{\mathbf{e}} = p_{\boldsymbol{\epsilon}}$ and $\boldsymbol{k}(X) = k\boldsymbol{k}'(X), k \to 0$. Assume further that **(D1)** and **(D2)** (Balance from data) $\mathcal{A}_0 = \mathcal{A}_1$ in* (5). *Then, if $p_{\boldsymbol{\theta}}(\boldsymbol{y}|\boldsymbol{x},t) = p(\boldsymbol{y}|\boldsymbol{x},t)$; conclusions 1) and 2) in Proposition 2 hold with $\mathfrak{p}$ replaced with $\mathfrak{p}_t$ in **(G1')**; and the domain of $\boldsymbol{v}$ becomes $\mathcal{P} \coloneqq \{\mathfrak{p}_t(\boldsymbol{x})|p(t,\boldsymbol{x}) > 0\}$.*

Theorem 1 implies that, without bPGS, we need to know or learn the distribution of hidden noise $\boldsymbol{\epsilon}$ to have $p_{\mathbf{e}} = p_{\boldsymbol{\epsilon}}$. Proposition 2 and Theorem 1 achieve recovery and identification in a complementary manner; the former starts from the prior by $\mathfrak{p}_0 = \mathfrak{p}_1$ and $\boldsymbol{h}_0 = \boldsymbol{h}_1$, while the latter starts from the decoder by $\mathcal{A}_0 = \mathcal{A}_1$ and $p_{\mathbf{e}} = p_{\boldsymbol{\epsilon}}$. We see that $\mathcal{A}_0 = \mathcal{A}_1$ acts as a kind of balance because it replaces $\mathfrak{p}_0 = \mathfrak{p}_1$ in Proposition 2. We show in Sec. A a sufficient and necessary condition **(D2')** on data that ensures $\mathcal{A}_0 = \mathcal{A}_1$. Note that the singularities due to $k \to 0$ (e.g., $\boldsymbol{\lambda} \to \boldsymbol{0}$) cancel out in (5). See Sec. C.4 for more on the complementarity between the two identifications.

## 4 ESTIMATION BY $\beta$-INTACT-VAE

### 4.1 PRIOR AS BPGS, POSTERIOR AS PGS, AND $\beta$ AS REGULARIZATION STRENGTH

In Sec. 3.2, we see that the existence of bPGS (Proposition 2) is preferable in identifying the true DGP up to an equivalent expression—while Theorem 1 allows us to deal with PGS by adding other conditions. In learning our model with data, we formally require **(G1)** and further expect that **(G2)** holds approximately; the latter is true when $\boldsymbol{f}_t^*$ is injective and $\mathfrak{m}_0 \approx \mathfrak{m}_1$ ($\mathfrak{m}_t(X)$ is an approximate bPGS). Instead of the trivial regression $\mu_t(X) = \mathbb{E}(Y|X, T = t)$, we want to recover the approximate bPGS $\mathfrak{m}_t(X)$. This idea is common in practice. For example, in a real-world nutrition study (Huang & Chan, 2017), a reduction of 11 covariates recovers a 1-dimensional linear bPGS.

We consider two ways to recover an approximate bPGS by a VAE. One is to use a prior which does not depend on $t$, indicating a preference for bPGS. Namely, we set $\boldsymbol{\lambda}_0 = \boldsymbol{\lambda}_1$, denote $\boldsymbol{\Lambda}(X) \coloneqq \boldsymbol{\lambda}(X)$ and have $p_{\boldsymbol{\Lambda}}(\boldsymbol{z}|\boldsymbol{x})$ as the prior in (3). The decoder and encoder are factorized Gaussians:

$$p_{\boldsymbol{f},\boldsymbol{g}}(\boldsymbol{y}|\boldsymbol{z},t) = \mathcal{N}(\boldsymbol{y}; \boldsymbol{f}_t(\boldsymbol{z}), \mathrm{diag}(\boldsymbol{g}_t(\boldsymbol{z}))), \quad q_{\boldsymbol{\phi}}(\boldsymbol{z}|\boldsymbol{x},\boldsymbol{y},t) = \mathcal{N}(\boldsymbol{z}; \boldsymbol{r}_t(\boldsymbol{x},\boldsymbol{y}), \mathrm{diag}(\boldsymbol{s}_t(\boldsymbol{x},\boldsymbol{y}))), \quad (6)$$

where $\boldsymbol{\phi} = (\boldsymbol{r}, \boldsymbol{s})$. The other is to introduce a hyperparameter $\beta$ in the ELBO as in $\beta$-VAE (Higgins et al., 2017). The modified ELBO with $\beta$, up to the additive constant, is derived as:

$$\mathbb{E}_{\mathcal{D}}\{-\beta D_{\mathrm{KL}}(q_{\boldsymbol{\phi}}\|p_{\boldsymbol{\Lambda}}) - \mathbb{E}_{\boldsymbol{z} \sim q_{\boldsymbol{\phi}}}[(\boldsymbol{y} - \boldsymbol{f}_t(\boldsymbol{z}))^2/2\boldsymbol{g}_t(\boldsymbol{z})] - \mathbb{E}_{\boldsymbol{z} \sim q_{\boldsymbol{\phi}}} \log |\boldsymbol{g}_t(\boldsymbol{z})|\}. \quad (7)$$

For convenience, here and in $\mathcal{L}_{\boldsymbol{f}}$ in Sec. 4.2, we omit the summation as if $Y$ is univariate. The encoder $q_{\boldsymbol{\phi}}$ depends on $t$ and can realize a PGS. With $\beta$, we control the trade-off between the first and second terms: the former is the divergence of the posterior from the balanced prior, and the latter is the reconstruction of the outcome. Note that a larger $\beta$ encourages the conditional

balance $Z \perp\!\!\!\perp T | X$ on the posterior. By choosing $\beta$ appropriately, e.g., by validation, the ELBO can recover an approximate bPGS while fitting the outcome well. In summary, we base the estimation on Proposition 2 and bPGS as much as possible, but step into Theorem 1 and noise modeling required by $p_{\mathbf{e}} = p_\epsilon$ when necessary.

Note also that the parameters $\boldsymbol{g}$ and $\boldsymbol{k}$, which model the outcome noise and express the uncertainty of the prior, respectively, are both learned by the ELBO. This deviates from the theoretical conditions described in Sec. 3.2, but it is more practical and yields better results in our experiments. See Sec. C.5 for more ideas and connections behind the ELBO.

Once the VAE is learned[2] by the ELBO, the estimate of the expected potential outcomes is given by:

$$\hat{\mu}_{\hat{t}}(\boldsymbol{x}) = \mathbb{E}_{q(\boldsymbol{z}|\boldsymbol{x})} \boldsymbol{f}_{\hat{t}}(\boldsymbol{z}) = \mathbb{E}_{\mathcal{D}|\boldsymbol{x} \sim p(\boldsymbol{y}, t|\boldsymbol{x})} \mathbb{E}_{\boldsymbol{z} \sim q_\phi} \boldsymbol{f}_{\hat{t}}(\boldsymbol{z}), \ \hat{t} \in \{0, 1\}, \tag{8}$$

where $q(\boldsymbol{z}|\boldsymbol{x}) \coloneqq \mathbb{E}_{p(\boldsymbol{y}, t|\boldsymbol{x})} q_\phi(\boldsymbol{z}|\boldsymbol{x}, \boldsymbol{y}, t)$ is the aggregated posterior. We mainly consider the case where $\boldsymbol{x}$ is observed in the data, and the sample of $(Y, T)$ is taken from the data given $X = \boldsymbol{x}$. When $\boldsymbol{x}$ is not in the data, we replace $q_\phi$ with $p_{\boldsymbol{\Lambda}}$ in (8) (see Sec. C.7 for details and E for results). Note that $\hat{t}$ in (8) indicates a counterfactual assignment that may not be the same as the factual $T = t$ in the data. That is, we set $T = \hat{t}$ in the decoder. The assignment is not applied to the encoder which is learned from factual $X, Y, T$ (see also the explanation of $\epsilon_{CF,t}$ in Sec. 4.2). The overall **algorithm** steps are i) train the VAE using (7), and ii) infer CATE $\hat{\tau}(\boldsymbol{x}) = \hat{\mu}_1(\boldsymbol{x}) - \hat{\mu}_0(\boldsymbol{x})$ by (8).

## 4.2 CONDITIONALLY BALANCED REPRESENTATION LEARNING

We formally justify our ELBO (7) from the BRL viewpoint. We show that the conditional BRL via the KL (first) term of the ELBO results from bounding a CATE error; particularly, the error due to the imprecise recovery of $\boldsymbol{j}_t$ in **(G1')** is controlled by the ELBO. Previous works (Shalit et al., 2017; Lu et al., 2020) instead focus on unconditional balance and bound PEHE which is marginalized on $X$. Sec. 5.2 experimentally shows the advantage of our bounds and ELBO. Further, we connect the bounds to identification and consider noise modeling through $\boldsymbol{g}_t(\boldsymbol{z})$. See Sec. D.3 for detailed comparisons to previous works. In Sec. E.4, we empirically validate our bounds, and, particularly, the bounds are more useful under weaker overlap.

We introduce the objective that we bound. Using (8) to estimate CATE, $\hat{\tau}_{\boldsymbol{f}}(\boldsymbol{z}) \coloneqq \boldsymbol{f}_1(\boldsymbol{z}) - \boldsymbol{f}_0(\boldsymbol{z})$ is marginalized on $q(\boldsymbol{z}|\boldsymbol{x})$. On the other hand, the *true CATE*, given the covariate $\boldsymbol{x}$ or score $\boldsymbol{z}$, is:

$$\tau(\boldsymbol{x}) = \boldsymbol{j}_1(\mathfrak{p}_1(\boldsymbol{x})) - \boldsymbol{j}_0(\mathfrak{p}_0(\boldsymbol{x})), \quad \tau_{\boldsymbol{j}}(\boldsymbol{z}) = \boldsymbol{j}_1(\boldsymbol{z}) - \boldsymbol{j}_0(\boldsymbol{z}), \tag{9}$$

where $\boldsymbol{j}_t$ is associated with an approximate bPGS $\mathfrak{p}_t$ (say, $\mathfrak{m}_t$) as the target of recovery by our VAE. Accordingly, given $\boldsymbol{x}$, the *error of posterior CATE*, with or without knowing $\mathfrak{p}_t$, is defined as

$$\epsilon_{\boldsymbol{f}}^*(\boldsymbol{x}) \coloneqq \mathbb{E}_{q(\boldsymbol{z}|\boldsymbol{x})}(\hat{\tau}_{\boldsymbol{f}}(\boldsymbol{z}) - \tau(\boldsymbol{x}))^2; \quad \epsilon_{\boldsymbol{f}}(\boldsymbol{x}) \coloneqq \mathbb{E}_{q(\boldsymbol{z}|\boldsymbol{x})}(\hat{\tau}_{\boldsymbol{f}}(\boldsymbol{z}) - \tau_{\boldsymbol{j}}(\boldsymbol{z}))^2. \tag{10}$$

We bound $\epsilon_{\boldsymbol{f}}$ instead of $\epsilon_{\boldsymbol{f}}^*$ because the error between $\tau(X)$ and $\tau_{\boldsymbol{j}}(Z)$ is small—if the score recovery works well, then $\boldsymbol{z} \approx \mathfrak{p}_0(\boldsymbol{x}) \approx \mathfrak{p}_1(\boldsymbol{x})$ in (9). We consider the error between $\hat{\tau}_{\boldsymbol{f}}$ and $\tau_{\boldsymbol{j}}$ below. We define the risks of outcome regression, into which $\epsilon_{\boldsymbol{f}}$ is decomposed.

**Definition 3** (CATE risks). Let $Y(\hat{t})|\mathfrak{p}_{\hat{t}}(X) \sim p_{Y(\hat{t})|\mathfrak{p}_{\hat{t}}}(\boldsymbol{y}|P)$ and $q_t(\boldsymbol{z}|\boldsymbol{x}) \coloneqq q(\boldsymbol{z}|\boldsymbol{x}, t) = \mathbb{E}_{p(\boldsymbol{y}|\boldsymbol{x}, t)} q_\phi$. The *potential outcome loss* at $(\boldsymbol{z}, t)$, *factual risk*, and *counterfactual risk* are:

$$\mathcal{L}_{\boldsymbol{f}}(\boldsymbol{z}, \hat{t}) \coloneqq \mathbb{E}_{p_{Y(\hat{t})|\mathfrak{p}_{\hat{t}}}(\boldsymbol{y}|P=\boldsymbol{z})}(\boldsymbol{y} - \boldsymbol{f}_{\hat{t}}(\boldsymbol{z}))^2 / \boldsymbol{g}_{\hat{t}}(\boldsymbol{z}) = \boldsymbol{g}_{\hat{t}}(\boldsymbol{z})^{-1} \int (\boldsymbol{y} - \boldsymbol{f}_{\hat{t}}(\boldsymbol{z}))^2 p_{Y(\hat{t})|\mathfrak{p}_{\hat{t}}}(\boldsymbol{y}|\boldsymbol{z}) d\boldsymbol{y};$$

$$\epsilon_{F,t}(\boldsymbol{x}) \coloneqq \mathbb{E}_{q_t(\boldsymbol{z}|\boldsymbol{x})} \mathcal{L}_{\boldsymbol{f}}(\boldsymbol{z}, t); \quad \epsilon_{CF,t}(\boldsymbol{x}) \coloneqq \mathbb{E}_{q_{1-t}(\boldsymbol{z}|\boldsymbol{x})} \mathcal{L}_{\boldsymbol{f}}(\boldsymbol{z}, t).$$

With $Y(t)$ involved, $\mathcal{L}_{\boldsymbol{f}}$ is a potential outcome loss on $\boldsymbol{f}$, weighted by $\boldsymbol{g}$. The factual and counterfactual counterparts, $\epsilon_{F,t}$ and $\epsilon_{CF,t}$, are defined accordingly. In $\epsilon_{F,t}$, unit $\boldsymbol{u} = (\boldsymbol{x}, \boldsymbol{y}, t)$ is involved in the learning of $q_t(\boldsymbol{z}|\boldsymbol{x})$, as well as in $\mathcal{L}_{\boldsymbol{f}}(\boldsymbol{z}, t)$ since $Y(t) = \boldsymbol{y}$ for the unit. In $\epsilon_{CF,t}$, however, unit $\boldsymbol{u}' = (\boldsymbol{x}, \boldsymbol{y}', 1 - t)$ is involved in $q_{1-t}(\boldsymbol{z}|\boldsymbol{x})$, but not in $\mathcal{L}_{\boldsymbol{f}}(\boldsymbol{z}, t)$ since $Y(t) \neq \boldsymbol{y}' = Y(1 - t)$.

Thus, *the regression error (second) term in ELBO* (7) *controls* $\epsilon_{F,t}$ *via factual data*. On the other hand, $\epsilon_{CF,t}$ is not estimable due to the unobservable $Y(1 - T)$, but is bounded by $\epsilon_{F,t}$ plus $M\boldsymbol{D}(\boldsymbol{x})$ in Theorem 2 below—which, in turn, bounds $\epsilon_{\boldsymbol{f}}$ by decomposing it to $\epsilon_{F,t}$, $\epsilon_{CF,t}$, and $\mathbf{V}_Y$.

---

[2]As usual, we expect the variational inference and optimization procedure to be (near) optimal; that is, consistency of VAE. *Consistent estimation* using the prior is a direct corollary of the consistent VAE. See Sec. C.6 for formal statements and proofs. Under Gaussian models, it is possible to prove the consistency of the posterior estimation, as shown in Bonhomme & Weidner (2021).

**Theorem 2** (CATE error bound). *Assume $|\mathcal{L}_f(z, t)| \le M$ and $|g_t(z)| \le G$, then:*

$$\epsilon_f(x) \le 2[G(\epsilon_{F,0}(x) + \epsilon_{F,1}(x) + MD(x)) - V_Y(x)] \tag{11}$$

*where $D(x) := \sum_t \sqrt{D_{KL}(q_t \| q_{1-t})/2}$, and $V_Y(x) := \mathbb{E}_{q(z|x)} \sum_t \mathbb{E}_{p_{Y(t)|p_t}(y|z)} (y - j_t(z))^2$.*

$D(x)$ measures the imbalance between $q_t(z|x)$ and is symmetric for $t$. Correspondingly, *the KL term in ELBO* (7) *is symmetric for $t$ and balances $q_t(z|x)$ by encouraging $Z \perp\!\!\!\perp T | X$ for the posterior*. $V_Y(x)$ reflects the intrinsic variance in the DGP and can not be controlled. Estimating $G, M$ is nontrivial. Instead, we rely on $\beta$ in the ELBO (7) to weight the terms. We do not need two hyperparameters since $G$ *is implicitly controlled by the third term, a norm constraint, in ELBO*.

## 5 EXPERIMENTS

We compare our method with existing methods on three types of datasets. Here, we present two experiments; the remaining one on the Pokec dataset is deferred to Sec. E.3. As in previous works (Shalit et al., 2017; Louizos et al., 2017), we report the absolute error of ATE $\epsilon_{ate} := |\mathbb{E}_{\mathcal{D}}(y(1) - y(0)) - \mathbb{E}_{\mathcal{D}}\hat{\tau}(x)|$ and, as a surrogate of square CATE error $\epsilon_{cate}(x) = \mathbb{E}_{\mathcal{D}|x}[(y(1) - y(0)) - \hat{\tau}(x)]^2$, the empirical PEHE $\epsilon_{pehe} := \mathbb{E}_{\mathcal{D}}\epsilon_{cate}(x)$ (Hill, 2011), which is the average square CATE error.

Unless otherwise indicated, for each function $f, g, h, k, r, s$ in ELBO (7), we use a multilayer perceptron, with $200 * 3$ hidden units (width 200, 3 layers), and ELU activations (Clevert et al., 2015). $\Lambda = (h, k)$ depends only on $X$. The Adam optimizer with initial learning rate $10^{-4}$ and batch size 100 is employed. All experiments use early-stopping of training by evaluating the ELBO on a validation set. More details on hyper-parameters and settings are given in each experiment.

### 5.1 SYNTHETIC DATASET

$$W|X \sim \mathcal{N}(h(X), k(X)); \ T|X \sim \text{Bern}(\text{Logi}(\omega l(X))); \ Y|W, T \sim \mathcal{N}(f_T(W), g_T(W)). \tag{12}$$

We generate synthetic datasets following (12). Both $X \sim \mathcal{N}(\mu, \sigma)$ and $W$ are factorized Gaussians. $\mu, \sigma$ are randomly sampled. The functions $h, k, l$ are linear. Outcome models $f_0, f_1$ are built by NNs with invertible activations. $Y$ is univariate, $\dim(X) = 30$, and $\dim(W)$ ranges from 1 to 5. $W$ is a bPGS, but the dimensionality is not low enough to satisfy the injectivity in **(G2)**, when $\dim(W) > 1$. We have 5 different overlap levels controlled by $\omega$ that multiplies the logit value. See Sec. E.1 for details and more results on synthetic datasets.

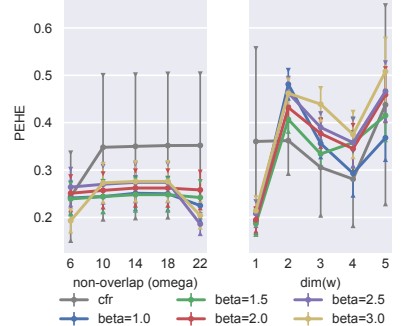

With the same $(\dim(W), \omega)$, we evaluate our method and CFR on 10 random DGPs, with different sets of functions $f, g, h, k, l$ in (12). For each DGP, we sample 1500 data points, and split them into 3 equal sets for training, valida-

Figure 2: $\sqrt{\epsilon_{pehe}}$ on synthetic datasets. Each error bar is on 10 random DGPs.

tion, and testing. We show our results for different hyperparameter $\beta$. For CFR, we try different balancing parameters and present the best results (see the Appendix for detail).

In each panel of Figure 2, we adjust one of $\omega, \dim(W)$, with the other fixed to the lowest. As implied by our theory, our method, with only *1-dimensional $Z$*, performs much better in the left panel (where $\dim(W) = 1$ satisfies **(G2)**) than in the right panel (when $\dim(W) > 1$). Although CFR uses 200-dimensional representation, in the left panel our method performs much better than CFR; moreover, in the right panel CFR is not much better than ours. Further, our

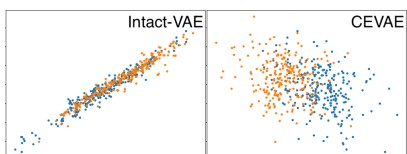

Figure 3: Plots of recovered - true latent. Blue: $T = 0$, Orange: $T = 1$.

method is much more robust against different DGPs than CFR (see the error bars). Thus, the results indicate the power of identification and recovery of scores. (see Figure 3 also).

Under the lowest overlap level ($\omega = 22$), large $\beta(= 2.5, 3)$ shows the best results, which accords with the intuition and bounds in Sec. 4. When $\dim(W) > 1$, $f_t$ in (12) is non-injecitve and learning

of PGS is necessary, and thus, larger $\beta$ has a negative effect. In fact, $\beta = 1$ is significantly better than $\beta = 3$ when $\dim(W) > 2$. We note that our method, with a higher-dimensional $Z$, outperforms or matches CFR also under $\dim(W) > 1$ (see Appendix Figure 5). Thus, the performance gap under $\dim(W) > 1$ in Figure 2 should be due to the capacity of NNs in $\beta$-Intact-VAE. In Appendix Figure 7 for ATE error, CFR drops performance w.r.t overlap levels. This is evidence that CFR and its unconditional balance overly focus on PEHE (see Sec. 5.2 for more explicit comparison).

When $\dim(W) = 1$, there are no better PSs than $W$, because $f_t$ is invertible and no information can be dropped from $W$. Thus, our method stably learns $Z$ as an approximate affine transformation of the true $W$, showing identification. An example is shown in Figure 3, and more plots are in Appendix Figure 9. For comparison, we run CEVAE, which is also based on VAE but without identification; CEVAE shows much lower quality of recovery. As expected, both recovery and estimation are better with the balanced prior $p_{\boldsymbol{\Lambda}}(\boldsymbol{z}|\boldsymbol{x})$, and we can see examples of bad recovery using $p_{\boldsymbol{\lambda}}(\boldsymbol{z}|\boldsymbol{x}, t)$ in Appendix Figure 10.

## 5.2 IHDP benchmark dataset

This experiment shows our conditional BRL matches state-of-the-art BRL methods and does not overly focus on PEHE. The IHDP (Hill, 2011) is a widely used benchmark dataset; while it is less known, its covariates are limited-overlapping, and thus it is used in Johansson et al. (2020) which considers limited overlap. The dataset is based on an RCT, but `Race` is artificially introduced as a confounder by removing all treated babies with nonwhite mothers in the data. Thus, `Race` is highly limited-overlapping, and other covariates that have high correlation to `Race`, e.g, `Birth weight` (Kelly et al., 2009), are also limited-overlapping. See Sec. E.2 for detail and more results.

There is a linear bPGS (linear combination of the covariates). However, most of the covariates are binary, so the support of the bPGS is often on small and separated intervals. Thus, the Gaussian latent $Z$ in our model is misspecified. We use higher-dimensional $Z$ to address this, similar to Louizos et al. (2017). Specifically, we set $\dim(Z) = 50$, together with NNs of $50 * 2$ hidden units in the prior and encoder. We set $\beta = 1$ since it works well on synthetic datasets with limited overlap.

As shown in Table 1, $\beta$-Intact-VAE outperforms or matches the state-of-the-art methods; it has the best performance measured by both $\epsilon_{ate}$ and $\epsilon_{pehe}$ and matches CF and CFR respectively. Also notably, our method outperforms other generative models (CEVAE and GANITE) by large margins.

To show our conditional balance is preferable, we also modify our method and add two components for *unconditional* balance from CFR (see the Appendix), which is based on bounding PEHE and is controlled by another hyperparameter $\gamma$. In the modified version, the over-focus on PEHE of the unconditional balance is seen clearly–with different $\gamma$, it significantly affects PEHE, but barely affects ATE error. In fact, the unconditional balance, with larger $\gamma$, only worsens the performance. See also Appendix Figure 7 where CFR gives larger ATE errors with less overlap.

Table 1: Errors on IHDP over 1000 random DGPs. "Mod. *" indicates the modified version with unconditional balance of strength $\gamma = *$. *Italic* indicates where the modified version is significantly worse than the original. **Bold** indicates method(s) which is significantly better than others. The results of other methods are taken from Shalit et al. (2017), except for GANITE and CEVAE, the results of which are taken from original works.

| Method | TMLE | BNN | CFR | CF | CEVAE | GANITE | Ours | Mod. 1 | Mod. 0.2 | Mod. 0.1 | Mod. 0.05 | Mod. 0.01 |
|---|---|---|---|---|---|---|---|---|---|---|---|---|
| $\epsilon_{ate}$ | $.30_{\pm.01}$ | $.37_{\pm.03}$ | $.25_{\pm.01}$ | $\mathbf{.18}_{\pm.01}$ | $.34_{\pm.01}$ | $.43_{\pm.05}$ | $\mathbf{.180}_{\pm.007}$ | $.185_{\pm.008}$ | $.185_{\pm.008}$ | $.186_{\pm.009}$ | $.183_{\pm.008}$ | $.181_{\pm.008}$ |
| $\sqrt{\epsilon_{pehe}}$ | $5.0_{\pm.2}$ | $2.2_{\pm.1}$ | $\mathbf{.71}_{\pm.02}$ | $3.8_{\pm.2}$ | $2.7_{\pm.1}$ | $1.9_{\pm.4}$ | $\mathbf{.709}_{\pm.024}$ | $\mathit{1.175}_{\pm.046}$ | $.797_{\pm.030}$ | $.748_{\pm.028}$ | $.732_{\pm.028}$ | $.719_{\pm.027}$ |

## 6 Conclusion

We proposed a method for CATE estimation under limited overlap. Our method exploits identifiable VAE, a recent advance in generative models, and is fully motivated and theoretically justified by causal considerations: identification, prognostic score, and balance. Experiments show evidence that the injectivity of $\boldsymbol{f}_t$ in our model is possibly unnecessary because $\dim(Z) > \dim(Y)$ yields better results. A theoretical study of this is an interesting future direction. We have evidence that Intact-VAE works under unobserved confounding and believe that VAEs are suitable for *principled* causal inference owing to their probabilistic nature, if not compromised by ad hoc heuristics (Wu & Fukumizu, 2021).

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

## A    PROOFS

We restate our model identifiability formally.

**Lemma 1** (Model identifiability). *Given model* (3) *under (M1), for $T = t$, assume*

**(D1')** (Non-degenerated data for $\boldsymbol{\lambda}$) there exist $2n + 1$ points $\boldsymbol{x}_0, ..., \boldsymbol{x}_{2n} \in \mathcal{X}$ such that the $2n$-square matrix $\boldsymbol{L}_t := [\boldsymbol{\gamma}_{t,1}, ..., \boldsymbol{\gamma}_{t,2n}]$ is invertible, where $\boldsymbol{\gamma}_{t,k} := \boldsymbol{\lambda}_t(\boldsymbol{x}_k) - \boldsymbol{\lambda}_t(\boldsymbol{x}_0)$.

*Then, given $T = t$, the family is* identifiable *up to an equivalence class. That is, if $p_{\boldsymbol{\theta}}(\boldsymbol{y}|\boldsymbol{x}, t) = p_{\boldsymbol{\theta}'}(\boldsymbol{y}|\boldsymbol{x}, t)$, we have the relation between parameters: for any $\boldsymbol{y}_t$ in the image of $\boldsymbol{f}_t$,*

$$\boldsymbol{f}_t^{-1}(\boldsymbol{y}_t) = \mathrm{diag}(\boldsymbol{a})\boldsymbol{f}_t'^{-1}(\boldsymbol{y}_t) + \boldsymbol{b} =: \mathcal{A}_t(\boldsymbol{f}_t'^{-1}(\boldsymbol{y}_t)) \tag{13}$$

*where $\mathrm{diag}(\boldsymbol{a})$ is an invertible $n$-diagonal matrix and $\boldsymbol{b}$ is a $n$-vector, both depend on $\boldsymbol{\lambda}_t$ and $\boldsymbol{\lambda}_t'$.*

Note, **(D1)** in the main text implies **(D1')**, see Sec. B.2.3 in Khemakhem et al. (2020a). The main part of our model identifiability is essentially the same as that of Theorem 1 in Khemakhem et al. (2020a), but now adapted to include the dependency on $t$. Here we give an outline of the proof, and the details can be easily filled by referring to Khemakhem et al. (2020a). In the proof, subscripts $t$ are omitted for convenience.

*Proof of Lemma 1.*    Using **(M1)** i) and ii) , we transform $p_{\boldsymbol{f},\boldsymbol{\lambda}}(\boldsymbol{y}|\boldsymbol{x}, t) = p_{\boldsymbol{f}',\boldsymbol{\lambda}'}(\boldsymbol{y}|\boldsymbol{x}, t)$ into equality of noiseless distributions, that is,

$$q_{\boldsymbol{f}',\boldsymbol{\lambda}'}(\boldsymbol{y}) = q_{\boldsymbol{f},\boldsymbol{\lambda}}(\boldsymbol{y}) := p_{\boldsymbol{\lambda}}(\boldsymbol{f}^{-1}(\boldsymbol{y})|\boldsymbol{x}, t)vol(\boldsymbol{J}_{\boldsymbol{f}^{-1}}(\boldsymbol{y}))\mathbb{I}_{\mathcal{Y}}(\boldsymbol{y}) \tag{14}$$

where $p_{\boldsymbol{\lambda}}$ is the Gaussian density function of the conditional prior defined in (3) and $vol(A) := \sqrt{\det AA^T}$. $q_{\boldsymbol{f}',\boldsymbol{\lambda}'}$ is defined similarly to $q_{\boldsymbol{f},\boldsymbol{\lambda}}$.

Then, apply model (3) to (14), plug the $2n + 1$ points from **(D1')** into it, and re-arrange the resulting $2n + 1$ equations in matrix form, we have

$$\mathcal{F}'(Y) = \mathcal{F}(Y) := \boldsymbol{L}^T \boldsymbol{t}(\boldsymbol{f}^{-1}(Y)) - \boldsymbol{\beta} \tag{15}$$

where $\boldsymbol{t}(Z) := (Z, Z^2)^T$ is the sufficient statistics of factorized Gaussian, and $\boldsymbol{\beta}_t := (\alpha_t(\boldsymbol{x}_1) - \alpha_t(\boldsymbol{x}_0), ..., \alpha_t(\boldsymbol{x}_{2n}) - \alpha_t(\boldsymbol{x}_0))^T$ where $\alpha_t(X; \boldsymbol{\lambda}_t)$ is the log-partition function of the conditional prior in (3). $\mathcal{F}'$ is defined similarly to $\mathcal{F}$, but with $\boldsymbol{f}', \boldsymbol{\lambda}', \alpha'$

Since $\boldsymbol{L}$ is invertible, we have

$$\boldsymbol{t}(\boldsymbol{f}^{-1}(Y)) = \boldsymbol{A}\boldsymbol{t}(\boldsymbol{f}'^{-1}(Y)) + \boldsymbol{c} \tag{16}$$

where $\boldsymbol{A} = \boldsymbol{L}^{-T}\boldsymbol{L}'^T$ and $\boldsymbol{c} = \boldsymbol{L}^{-T}(\boldsymbol{\beta} - \boldsymbol{\beta}')$.

The final part of the proof is to show, by following the same reasoning as in Appendix B of Sorrenson et al. (2019), that $\boldsymbol{A}$ is a sparse matrix such that

$$\boldsymbol{A} = \begin{pmatrix} \mathrm{diag}(\boldsymbol{a}) & \boldsymbol{O} \\ \mathrm{diag}(\boldsymbol{u}) & \mathrm{diag}(\boldsymbol{a}^2) \end{pmatrix} \tag{17}$$

where $\boldsymbol{A}$ is partitioned into four $n$-square matrices. Thus

$$\boldsymbol{f}^{-1}(Y) = \mathrm{diag}(\boldsymbol{a})\boldsymbol{f}'^{-1}(Y) + \boldsymbol{b} \tag{18}$$

where $\boldsymbol{b}$ is the first half of $\boldsymbol{c}$.    $\square$

*Proof of Proposition 2.*    Under **(G2)**, and **(M3)**, we have

$$\mathbb{E}_{p_{\boldsymbol{\theta}}}(Y|X, T) = \mathbb{E}(Y|X, T) \implies \boldsymbol{f}_t \circ \boldsymbol{h}(\boldsymbol{x}) = \boldsymbol{j}_t \circ \mathfrak{p}(\boldsymbol{x}) \text{ on } (\boldsymbol{x}, t) \text{ such that } p(t, \boldsymbol{x}) > 0. \tag{19}$$

We show the solution set of (19) on *overlapping $\boldsymbol{x}$* is

$$\{(\boldsymbol{f}, \boldsymbol{h})|\boldsymbol{f}_t = \boldsymbol{j}_t \circ \Delta^{-1}, \boldsymbol{h} = \Delta \circ \mathfrak{p}, \Delta : \mathcal{P} \to \mathbb{R}^n \text{ is injective}\}. \tag{20}$$

By **(G2)(M1)**, and with injective $\boldsymbol{f}_t, \boldsymbol{j}_t$ and $\dim(Z) = \dim(Y) \geq \dim(\mathfrak{p})$, for any $\Delta$ above, there exists a functional parameter $\boldsymbol{f}_t$ such that $\boldsymbol{j}_t = \boldsymbol{f}_t \circ \Delta$. Thus, set (20) is non-empty, and any element is indeed a solution because $\boldsymbol{f}_t \circ \boldsymbol{h} = \boldsymbol{j}_t \circ \Delta^{-1} \circ \Delta \circ \mathfrak{p} = \boldsymbol{j}_t \circ \mathfrak{p}$.

Any solution of (19) should be in (20). A solution should satisfy $\boldsymbol{h}(\boldsymbol{x}) = \boldsymbol{f}_t^{-1} \circ \boldsymbol{j}_t \circ \mathfrak{p}(\boldsymbol{x})$ for both $t$ since $\boldsymbol{x}$ is overlapping. This means the *injective* function $\boldsymbol{f}_t^{-1} \circ \boldsymbol{j}_t$ should *not* depend on $t$, thus it is one of the $\Delta$ in (20).

We proved conclusion 1) with $\boldsymbol{v} := \Delta$. And, on overlapping $\boldsymbol{x}$, conclusion 2) is quickly seen from

$$\hat{\mu}_t(\boldsymbol{x}) = \boldsymbol{f}_t(\boldsymbol{h}(\boldsymbol{x})) = \boldsymbol{j}_t \circ \boldsymbol{v}^{-1}(\boldsymbol{v} \circ \mathfrak{p}(\boldsymbol{x})) = \boldsymbol{j}_t(\mathfrak{p}(\boldsymbol{x})) = \mu_t(\boldsymbol{x}). \tag{21}$$

We rely on overlapping $\mathfrak{p}$ to work for non-overlapping $\boldsymbol{x}$. For any $\boldsymbol{x}_t$ with $p(1-t|\boldsymbol{x}_t) = 0$, to ensure $p(1-t|\mathfrak{p}(\boldsymbol{x}_t)) > 0$, there should exist $\boldsymbol{x}_{1-t}$ such that $\mathfrak{p}(\boldsymbol{x}_{1-t}) = \mathfrak{p}(\boldsymbol{x}_t)$ and $p(1-t|\boldsymbol{x}_{1-t}) > 0$. And we also have $\boldsymbol{h}(\boldsymbol{x}_{1-t}) = \boldsymbol{h}(\boldsymbol{x}_t)$ due to **(M2)**. Then, we have

$$\hat{\mu}_{1-t}(\boldsymbol{x}_t) = \boldsymbol{f}_{1-t}(\boldsymbol{h}(\boldsymbol{x}_t)) = \boldsymbol{f}_{1-t}(\boldsymbol{h}(\boldsymbol{x}_{1-t})) = \boldsymbol{j}_{1-t}(\mathfrak{p}(\boldsymbol{x}_{1-t})) = \boldsymbol{j}_{1-t}(\mathfrak{p}(\boldsymbol{x}_t)) = \mu_{1-t}(\boldsymbol{x}_t). \tag{22}$$

The third equality uses (19) on $(\boldsymbol{x}_{1-t}, 1-t)$. $\qquad\square$

Below we prove Theorem 1 with **(D2)** replaced by

**(D2')** *(Spontaneous balance) there exist $2n+1$ points $\boldsymbol{x}_0, ..., \boldsymbol{x}_{2n} \in \mathcal{X}$, $2n$-square matrix $\boldsymbol{C}$, and $2n$-vector $\boldsymbol{d}$, such that $\boldsymbol{L}_0^{-1}\boldsymbol{L}_1 = \boldsymbol{C}$ and $\boldsymbol{\beta}_0 - \boldsymbol{C}^{-T}\boldsymbol{\beta}_1 = \boldsymbol{d}/k$ for optimal $\boldsymbol{\lambda}_t$ (see below), where $\boldsymbol{L}_t$ is defined in (D1'), $\boldsymbol{\beta}_t := (\alpha_t(\boldsymbol{x}_1) - \alpha_t(\boldsymbol{x}_0), ..., \alpha_t(\boldsymbol{x}_{2n}) - \alpha_t(\boldsymbol{x}_0))^T$, and $\alpha_t(X; \boldsymbol{\lambda}_t)$ is the log-partition function of the prior in (3).*

**(D2')** restricts the discrepancy between $\boldsymbol{\lambda}_0, \boldsymbol{\lambda}_1$ on $2n+1$ values of $X$, thus is relatively easy to satisfy with high-dimensional $X$. **(D2')** is general despite (or thanks to) the involved formulation. Let us see its generality even under a highly special case: $\boldsymbol{C} = c\boldsymbol{I}$ and $\boldsymbol{d} = \boldsymbol{0}$. Then, $\boldsymbol{L}_0^{-1}\boldsymbol{L}_1 = c\boldsymbol{I}$ requires that, $\boldsymbol{h}_1(\boldsymbol{x}_k) - c\boldsymbol{h}_0(\boldsymbol{x}_k)$ is the same for $2n+1$ points $\boldsymbol{x}_k$. This is easily satisfied except for $n \gg m$ where $m$ is the dimension of $X$, which *rarely* happens in practice. And, $\boldsymbol{\beta}_0 - \boldsymbol{C}^{-T}\boldsymbol{\beta}_1 = \boldsymbol{d}$ becomes just $\boldsymbol{\beta}_1 = c\boldsymbol{\beta}_0$. This is equivalent to $\alpha_1(\boldsymbol{x}_k) - c\alpha_0(\boldsymbol{x}_k)$ same for $2n+1$ points, again fine in practice. However, the high generality comes with price. Verifying **(D2')** using data is challenging, particularly with high-dimensional covariate and latent variable. Although we believe fast algorithms for this purpose could be developed, the effort would be nontrivial. This is another motivation to use the extreme case $\boldsymbol{\lambda}_0 = \boldsymbol{\lambda}_1$ in Sec. 4.1, which corresponds to $\boldsymbol{C} = \boldsymbol{I}$ and $\boldsymbol{d} = \boldsymbol{0}$.

*Proof of Theorem 1.* By **(M1)** and **(G1')**, for any injective function $\Delta : \mathcal{P} \to \mathbb{R}^n$, there exists a functional parameter $\boldsymbol{f}_t^*$ such that $\boldsymbol{j}_t = \boldsymbol{f}_t^* \circ \Delta$. Let $\boldsymbol{h}_t^* = \Delta \circ \mathfrak{p}_t$, then, clearly from **(M3')**, such parameters $\boldsymbol{\theta}^* = (\boldsymbol{f}^*, \boldsymbol{h}^*)$ are optimal: $p_{\boldsymbol{\theta}^*}(\boldsymbol{y}|\boldsymbol{x}, t) = p(\boldsymbol{y}|\boldsymbol{x}, t)$.

Since have all assumptions for Lemma 1, we have

$$\Delta \circ \boldsymbol{j}^{-1}(\boldsymbol{y}) = \boldsymbol{f}^{*-1}(\boldsymbol{y}) = \mathcal{A} \circ \boldsymbol{f}^{-1}(\boldsymbol{y})|_t, \text{ on } (\boldsymbol{y}, t) \in \{(\boldsymbol{j}_t \circ \mathfrak{p}_t(\boldsymbol{x}), t)|p(t, \boldsymbol{x}) > 0\}, \tag{23}$$

where $\boldsymbol{f}$ is *any* optimal parameter, and "$|_t$" collects all subscripts $t$. Note, except for $\Delta$, all the symbols should have subscript $t$.

Nevertheless, using **(D2')**, we can further prove $\mathcal{A}_0 = \mathcal{A}_1$.

We repeat the core quantities from Lemma 1 here: $\boldsymbol{A}_t = \boldsymbol{L}_t^{-T}\boldsymbol{L}_t'^T$ and $\boldsymbol{c}_t = \boldsymbol{L}_t^{-T}(\boldsymbol{\beta}_t - \boldsymbol{\beta}_t')$.

From **(D2')**, we immediately have

$$\boldsymbol{L}_0^{-1}\boldsymbol{L}_1 = \boldsymbol{L}_0'^{-1}\boldsymbol{L}_1' = \boldsymbol{C} \iff \boldsymbol{A}_0 = \boldsymbol{A}_1 \tag{24}$$

And also,

$$\boldsymbol{L}_0^{-1}\boldsymbol{L}_1 = \boldsymbol{C} \iff \boldsymbol{L}_0^{-T}\boldsymbol{C}^{-T} = \boldsymbol{L}_1^{-T}$$
$$\boldsymbol{\beta}_0 - \boldsymbol{C}^{-T}\boldsymbol{\beta}_1 = \boldsymbol{\beta}_0' - \boldsymbol{C}^{-T}\boldsymbol{\beta}_1' = \boldsymbol{d}/k \iff \boldsymbol{C}^T(\boldsymbol{\beta}_0 - \boldsymbol{\beta}_0') = \boldsymbol{\beta}_1 - \boldsymbol{\beta}_1' \tag{25}$$

Multiply right hand sides of the two lines, we have $\boldsymbol{c}_0 = \boldsymbol{c}_1$. Now we have $\mathcal{A}_0 = \mathcal{A}_1 := \mathcal{A}$. Apply this to (23), we have

$$\boldsymbol{f}_t = \boldsymbol{j}_t \circ \boldsymbol{v}^{-1}, \quad \boldsymbol{v} := \mathcal{A}^{-1} \circ \Delta \tag{26}$$

for *any* optimal parameters $\boldsymbol{\theta} = (\boldsymbol{f}, \boldsymbol{h})$. Again, from **(M3')**, we have

$$p_{\boldsymbol{\theta}}(\boldsymbol{y}|\boldsymbol{x}, t) = p(\boldsymbol{y}|\boldsymbol{x}, t) \implies p_{\boldsymbol{\epsilon}}(\boldsymbol{y} - \boldsymbol{f}_t(\boldsymbol{h}_t(\boldsymbol{x}))) = p_{\mathbf{e}}(\boldsymbol{y} - \boldsymbol{j}_t(\mathfrak{p}_t(\boldsymbol{x}))) \tag{27}$$

where $p_{\boldsymbol{\epsilon}} = p_{\mathbf{e}}$. And the above is only possible when $\boldsymbol{f}_t \circ \boldsymbol{h}_t = \boldsymbol{j}_t \circ \mathfrak{p}_t$. Combined with $\boldsymbol{f}_t = \boldsymbol{j}_t \circ \boldsymbol{v}^{-1}$, we have conclusion 1).

And conclusion 2) follows from the same reasoning as Proposition 2, applied to both $\mathfrak{p}_0$ and $\mathfrak{p}_1$. $\quad\square$

Note, when multiplying the two lines of (25), the effects of $k \to 0$ cancel out, and $\boldsymbol{c}_t$ is finite and well-defined. Also, it is apparent from above proof that **(D2')** is a necessary and sufficient condition for $\mathcal{A}_0 = \mathcal{A}_1$, if other conditions of Theorem 1 are given.

Below, we prove the results in Sec. 4.2. The definitions and results work for the prior; simply *replace $q_t(\boldsymbol{x}|\boldsymbol{x})$ with $p_t(\boldsymbol{z}|\boldsymbol{x}) := p_{\boldsymbol{\lambda}}(\boldsymbol{z}|\boldsymbol{x}, t)$ in definitions and statements, and the proofs below hold as the same.* The dependence on $\boldsymbol{f}$ prevail, and the superscripts are omitted. The arguments $\boldsymbol{x}$ are sometimes also omitted.

**Lemma 2** (Counterfactual risk bound). *Assume $|\mathcal{L}_{\boldsymbol{f}}(\boldsymbol{z}, t)| \leq M$, we have*

$$\epsilon_{CF}(\boldsymbol{x}) \leq \sum_t q(1 - t|\boldsymbol{x})\epsilon_{F,t}(\boldsymbol{x}) + M\boldsymbol{D}(\boldsymbol{x}) \tag{28}$$

*where $\epsilon_{CF}(\boldsymbol{x}) := \sum_t p(1 - t|\boldsymbol{x})\epsilon_{CF,t}(\boldsymbol{x})$, and $\boldsymbol{D}(\boldsymbol{x}) := \sum_t \sqrt{D_{\mathrm{KL}}(q_t\|q_{1-t})/2}$.*

*Proof of Lemma 2.*

$$\begin{aligned}
&\epsilon_{CF} - \sum_t p(1 - t|\boldsymbol{x})\epsilon_{F,t} \\
&= p(0|\boldsymbol{x})(\epsilon_{CF,1} - \epsilon_{F,1}) + p(1|\boldsymbol{x})(\epsilon_{CF,0} - \epsilon_{F,0}) \\
&= p(0|\boldsymbol{x}) \int \mathcal{L}_{\boldsymbol{f}}(\boldsymbol{z}, 1)(q_0(\boldsymbol{z}|\boldsymbol{x}) - q_1(\boldsymbol{z}|\boldsymbol{x}))d\boldsymbol{z} + p(1|\boldsymbol{x}) \int \mathcal{L}_{\boldsymbol{f}}(\boldsymbol{z}, 0)(q_1(\boldsymbol{z}|\boldsymbol{x}) - q_0(\boldsymbol{z}|\boldsymbol{x}))d\boldsymbol{z} \\
&\leq 2M\mathbb{TV}(q_1, q_0) \leq M\boldsymbol{D}.
\end{aligned}$$

$\square$

$\mathbb{TV}(p, q) := \frac{1}{2}\mathbb{E}|p(\boldsymbol{z}) - q(\boldsymbol{z})|$ is the total variance distance between probability density $p, q$. The last inequality uses Pinsker's inequality $\mathbb{TV}(p, q) \leq \sqrt{D_{\mathrm{KL}}(p\|q)/2}$ twice, to get the symmetric $\boldsymbol{D}$.

Theorem 2 is a direct corollary of Lemma 2 and the following.

**Lemma 3.** *Define $\epsilon_F = \sum_t p(t|\boldsymbol{x})\epsilon_{F,t}$. We have*

$$\epsilon_{\boldsymbol{f}} \leq 2(G(\epsilon_F + \epsilon_{CF}) - \mathbf{V}_Y). \tag{29}$$

*Simply bound $\epsilon_{CF}$ in (29) by Lemma 2, we have Theorem 2.* To prove Lemma 3, we first examine a bias-variance decomposition of $\epsilon_F$ and $\epsilon_{CF}$.

$$\begin{aligned}
\epsilon_{CF,t} &= \mathbb{E}_{q_{1-t}(\boldsymbol{z}|\boldsymbol{x})}\boldsymbol{g}_t(\boldsymbol{z})\mathbb{E}_{p_{Y(t)|\mathbf{p}_t}(\boldsymbol{y}|\boldsymbol{z})}(\boldsymbol{y} - \boldsymbol{f}_t(\boldsymbol{z}))^2 \\
&\geq G\mathbb{E}_{q_{1-t}(\boldsymbol{z}|\boldsymbol{x})}\mathbb{E}_{p_{Y(t)|\mathbf{p}_t}(\boldsymbol{y}|\boldsymbol{z})}(\boldsymbol{y} - \boldsymbol{f}_t(\boldsymbol{z}))^2 \\
&= G\mathbb{E}_{q_{1-t}(\boldsymbol{z}|\boldsymbol{x})}\mathbb{E}_{p_{Y(t)|\mathbf{p}_t}(\boldsymbol{y}|\boldsymbol{z})}((\boldsymbol{y} - \boldsymbol{j}_t(\boldsymbol{z}))^2 + (\boldsymbol{j}_t(\boldsymbol{z}) - \boldsymbol{f}_t(\boldsymbol{z}))^2)
\end{aligned} \tag{30}$$

The second line uses $|\boldsymbol{g}_t(\boldsymbol{z})| \leq G$, and the third line is a bias-variance decomposition. Now we can define $\mathbf{V}_{CF,t}(\boldsymbol{x}) := \mathbb{E}_{q_{1-t}(\boldsymbol{z}|\boldsymbol{x})}\mathbb{E}_{p_{Y(t)|\mathbf{p}_t}(\boldsymbol{y}|\boldsymbol{z})}(\boldsymbol{y} - \boldsymbol{j}_t(\boldsymbol{z}))^2$ and $\mathbb{B}_{CF,t}(\boldsymbol{x}) := \mathbb{E}_{q_{1-t}(\boldsymbol{z}|\boldsymbol{x})}(\boldsymbol{j}_t(\boldsymbol{z}) - \boldsymbol{f}_t(\boldsymbol{z}))^2$, and we have

$$\epsilon_{CF,t} \geq G(\mathbf{V}_{CF,t}(\boldsymbol{x}) + \mathbb{B}_{CF,t}(\boldsymbol{x})) \implies \epsilon_{CF} \geq G(\mathbf{V}_{CF}(\boldsymbol{x}) + \mathbb{B}_{CF}(\boldsymbol{x})) \tag{31}$$

where $\mathbf{V}_{CF} := \sum_t p(1 - t|\boldsymbol{x})\mathbf{V}_{CF,t} = \sum_t \mathbb{E}_{q(\boldsymbol{z}, 1-t|\boldsymbol{x})}\mathbb{E}_{p_{Y(t)|\mathbf{p}_t}(\boldsymbol{y}|\boldsymbol{z})}(\boldsymbol{y} - \boldsymbol{j}_t(\boldsymbol{z}))^2$ and similarly $\mathbb{B}_{CF} = \sum_t \mathbb{E}_{q(\boldsymbol{z}, 1-t|\boldsymbol{x})}(\boldsymbol{j}_t(\boldsymbol{z}) - \boldsymbol{f}_t(\boldsymbol{z}))^2$. Repeat the above derivation for $\epsilon_F$, we have

$$\epsilon_F \geq G(\mathbf{V}_F(\boldsymbol{x}) + \mathbb{B}_F(\boldsymbol{x})) \tag{32}$$

where $\mathbf{V}_F = \sum_t \mathbb{E}_{q(\boldsymbol{z}, t|\boldsymbol{x})}\mathbb{E}_{p_{Y(t)|\mathbf{p}_t}(\boldsymbol{y}|\boldsymbol{z})}(\boldsymbol{y} - \boldsymbol{j}_t(\boldsymbol{z}))^2$ and $\mathbb{B}_F = \sum_t \mathbb{E}_{q(\boldsymbol{z}, t|\boldsymbol{x})}(\boldsymbol{j}_t(\boldsymbol{z}) - \boldsymbol{f}_t(\boldsymbol{z}))^2$. Now, we are ready to prove Lemma 3.

*Proof of Lemma 3.*

$$\begin{aligned}
\epsilon_{\boldsymbol{f}} &= \mathbb{E}_{q(\boldsymbol{z}|\boldsymbol{x})}((\boldsymbol{f}_1 - \boldsymbol{f}_0) - (\boldsymbol{j}_1 - \boldsymbol{j}_0))^2 \\
&= \mathbb{E}_q((\boldsymbol{f}_1 - \boldsymbol{j}_1) + (\boldsymbol{j}_0 - \boldsymbol{f}_0))^2 \\
&\leq 2\mathbb{E}_q((\boldsymbol{f}_1 - \boldsymbol{j}_1)^2 + (\boldsymbol{j}_0 - \boldsymbol{f}_0)^2) \\
&= 2 \int [(\boldsymbol{f}_1 - \boldsymbol{j}_1)^2 q(\boldsymbol{z}, 1|\boldsymbol{x}) + (\boldsymbol{j}_0 - \boldsymbol{f}_0)^2 q(\boldsymbol{z}, 0|\boldsymbol{x}) + \\
&\qquad (\boldsymbol{f}_1 - \boldsymbol{j}_1)^2 q(\boldsymbol{z}, 0|\boldsymbol{x}) + (\boldsymbol{j}_0 - \boldsymbol{f}_0)^2 q(\boldsymbol{z}, 1|\boldsymbol{x})]d\boldsymbol{z} \\
&= 2(\mathbb{B}_F + \mathbb{B}_{CF}) \leq 2(G(\epsilon_F + \epsilon_{CF}) - \mathbf{V}_Y)
\end{aligned}$$

$\square$

The first inequality uses $(a + b)^2 \leq 2(a^2 + b^2)$. The next equality splits $q(\boldsymbol{z}|\boldsymbol{x})$ into $q(\boldsymbol{z}, 0|\boldsymbol{x})$ and $q(\boldsymbol{z}, 1|\boldsymbol{x})$ and rearranges to get $\mathbb{B}_F$ and $\mathbb{B}_{CF}$. The last inequality uses the two bias-variance decompositions, and $\mathbf{V}_Y = \mathbf{V}_F + \mathbf{V}_{CF}$.

# B  ADDITIONAL BACKGROUNDS

## B.1  PROGNOSTIC SCORE AND BALANCING SCORE

In the fundamental work of (Hansen, 2008), prognostic score is defined equivalently to our $\mathfrak{p}_0$ (P0-score), but it in addition requires no effect modification to work for $Y(1)$. Thus, a useful prognostic score corresponds to our PGS. We give main properties of PGS as following.

**Proposition 3.** *If $V$ gives exchangeability, and $\mathfrak{p}_t(V)$ is a PGS, then $Y(t) \perp\!\!\!\perp V, T | \mathfrak{p}_t$.*

The following three properties of conditional independence will be used repeatedly in proofs.

**Proposition 4** (Properties of conditional independence). *(Pearl, 2009, Sec. 1.1.55) For random variables $W, X, Y, Z$. We have:*

$$X \perp\!\!\!\perp Y | Z \wedge X \perp\!\!\!\perp W | Y, Z \implies X \perp\!\!\!\perp W, Y | Z \text{ (Contraction)}.$$
$$X \perp\!\!\!\perp W, Y | Z \implies X \perp\!\!\!\perp Y | W, Z \text{ (Weak union)}.$$
$$X \perp\!\!\!\perp W, Y | Z \implies X \perp\!\!\!\perp Y | Z \text{ (Decomposition)}.$$

*Proof of Proposition 3.* From $Y(t) \perp\!\!\!\perp T | V$ (*exchangeability* of $V$), and since $\mathfrak{p}_t$ is a *function* of $V$, we have $Y(t) \perp\!\!\!\perp T | \mathfrak{p}_t, V$ (1).

From (1) and $Y(t) \perp\!\!\!\perp V | \mathfrak{p}_t(V)$ (definition of Pt-score), using contraction rule, we have $Y(t) \perp\!\!\!\perp T, V | \mathfrak{p}_t$ for both $t$.  □

Prognostic scores are closely related to the important concept of balancing score (Rosenbaum & Rubin, 1983). Note particularly, the proposition implies $Y(t) \perp\!\!\!\perp T | \mathfrak{p}_t$ (using decomposition rule). Thus, if $\mathfrak{p}(V)$ is a P-score, then $\mathfrak{p}$ also gives weak ignorability (exchangeability and overlap), which is a nice property shared with balancing score, as we will see immediately.

**Definition 4** (Balancing score). *$\boldsymbol{b}(V)$, a function of random variable $V$, is a balancing score if $T \perp\!\!\!\perp V | \boldsymbol{b}(V)$.*

**Proposition 5.** *Let $\boldsymbol{b}(V)$ be a function of random variable $V$. $\boldsymbol{b}(V)$ is a balancing score if and only if $f(\boldsymbol{b}(V)) = p(T = 1|V) := e(V)$ for some function $f$ (or more formally, $e(V)$ is $\boldsymbol{b}(V)$-measurable). Assume further that $V$ gives weak ignorability, then so does $\boldsymbol{b}(V)$.*

Obviously, the *propensity score* $e(V) := p(T = 1|V)$, the propensity of assigning the treatment given $V$, is a balancing score (with $f$ be the identity function). Also, given any invertible function $\boldsymbol{v}$, the composition $\boldsymbol{v} \circ \boldsymbol{b}$ is also a balancing score since $f \circ \boldsymbol{v}^{-1}(\boldsymbol{v} \circ \boldsymbol{b}(V)) = f(\boldsymbol{b}(V)) = e(V)$.

Compare the definition of balancing score and prognostic score, we can say balancing score is sufficient for the treatment $T$ ($T \perp\!\!\!\perp V | \boldsymbol{b}(V)$), while prognostic score (Pt-score) is sufficient for the potential outcomes $Y(t)$ ($Y(t) \perp\!\!\!\perp V | \mathfrak{p}_t(V)$). They complement each other; conditioning on either deconfounds the potential outcomes from treatment, with the former focuses on the treatment side, the latter on the outcomes side.

## B.2  VAE, CONDITIONAL VAE, AND iVAE

VAEs (Kingma et al., 2019) are a class of latent variable models with latent variable $Z$, and observable $Y$ is generated by the decoder $p_{\boldsymbol{\theta}}(\boldsymbol{y}|\boldsymbol{z})$. In the standard formulation (Kingma & Welling, 2013), the variational lower bound $\mathcal{L}(\boldsymbol{y}; \boldsymbol{\theta}, \boldsymbol{\phi})$ of the log-likelihood is derived as:

$$\begin{aligned} \log p(\boldsymbol{y}) &\geq \log p(\boldsymbol{y}) - D_{\mathrm{KL}}(q(\boldsymbol{z}|\boldsymbol{y}) \| p(\boldsymbol{z}|\boldsymbol{y})) \\ &= \mathbb{E}_{\boldsymbol{z} \sim q} \log p_{\boldsymbol{\theta}}(\boldsymbol{y}|\boldsymbol{z}) - D_{\mathrm{KL}}(q_{\boldsymbol{\phi}}(\boldsymbol{z}|\boldsymbol{y}) \| p(\boldsymbol{z})), \end{aligned} \tag{33}$$

where $D_{\mathrm{KL}}$ denotes KL divergence and the encoder $q_{\boldsymbol{\phi}}(\boldsymbol{z}|\boldsymbol{y})$ is introduced to approximate the true posterior $p(\boldsymbol{z}|\boldsymbol{y})$. The decoder $p_{\boldsymbol{\theta}}$ and encoder $q_{\boldsymbol{\phi}}$ are usually parametrized by NNs. We will omit the parameters $\boldsymbol{\theta}, \boldsymbol{\phi}$ in notations when appropriate.

The parameters of the VAE can be learned with stochastic gradient variational Bayes. With Gaussian latent variables, the KL term of $\mathcal{L}$ has closed form, while the first term can be evaluated by drawing samples from the approximate posterior $q_\phi$ using the reparameterization trick (Kingma & Welling, 2013), then, optimizing the evidence lower bound (ELBO) $\mathbb{E}_{\boldsymbol{y} \sim \mathcal{D}}(\mathcal{L}(\boldsymbol{y}))$ with data $\mathcal{D}$, we train the VAE efficiently.

Conditional VAE (CVAE) (Sohn et al., 2015; Kingma et al., 2014) adds a conditioning variable $C$, usually a class label, to standard VAE (See Figure 1). With the conditioning variable, CVAE can give better reconstruction of each class. The variational lower bound is

$$\log p(\boldsymbol{y}|\boldsymbol{c}) \geq \mathbb{E}_{\boldsymbol{z} \sim q} \log p(\boldsymbol{y}|\boldsymbol{z}, \boldsymbol{c}) - D_{\mathrm{KL}}(q(\boldsymbol{z}|\boldsymbol{y}, \boldsymbol{c}) \| p(\boldsymbol{z}|\boldsymbol{c})). \tag{34}$$

The conditioning on $C$ in the prior is usually omitted (Doersch, 2016), i.e., the prior becomes $Z \sim \mathcal{N}(\boldsymbol{0}, \boldsymbol{I})$ as in standard VAE, since the dependence between $C$ and the latent representation is also modeled in the encoder $q$. Moreover, unconditional prior in fact gives better reconstruction because it encourages learning representation independent of class, similarly to the idea of beta-VAE (Higgins et al., 2017).

As mentioned, *identifiable* VAE (iVAE) (Khemakhem et al., 2020a) provides the first identifiability result for VAE, using auxiliary variable $X$. It assumes $Y \perp\!\!\!\perp X | Z$, that is, $p(\boldsymbol{y}|\boldsymbol{z}, \boldsymbol{x}) = p(\boldsymbol{y}|\boldsymbol{z})$. The variational lower bound is

$$\begin{aligned} \log p(\boldsymbol{y}|\boldsymbol{x}) &\geq \log p(\boldsymbol{y}|\boldsymbol{x}) - D_{\mathrm{KL}}(q(\boldsymbol{z}|\boldsymbol{y}, \boldsymbol{x}) \| p(\boldsymbol{z}|\boldsymbol{y}, \boldsymbol{x})) \\ &= \mathbb{E}_{\boldsymbol{z} \sim q} \log p_{\boldsymbol{f}}(\boldsymbol{y}|\boldsymbol{z}) - D_{\mathrm{KL}}(q(\boldsymbol{z}|\boldsymbol{y}, \boldsymbol{x}) \| p_{\boldsymbol{T}, \boldsymbol{\lambda}}(\boldsymbol{z}|\boldsymbol{x})), \end{aligned} \tag{35}$$

where $Y = \boldsymbol{f}(Z) + \boldsymbol{\epsilon}$, $\boldsymbol{\epsilon}$ is additive noise, and $Z$ has exponential family distribution with sufficient statistics $\boldsymbol{T}$ and parameter $\boldsymbol{\lambda}(X)$. Note that, unlike CVAE, the decoder does *not* depend on $X$ due to the independence assumption.

Here, *identifiability of the model* means that the functional *parameters* $(\boldsymbol{f}, \boldsymbol{T}, \boldsymbol{\lambda})$ can be identified (learned) up to certain simple transformation. Further, in the limit of $\boldsymbol{\epsilon} \to \boldsymbol{0}$, iVAE solves the nonlinear ICA problem of recovering $Z = \boldsymbol{f}^{-1}(Y)$.

## C EXPOSITIONS

The order of subsections below follows that they are referred in the main text.

### C.1 LIST OF ASSUMPTIONS

The following is a list of assumptions required by our identification theory, with comments on their roles and subtleties.

**(G1)** additive noise model is needed to ensure the existence of PtSs. **(G1')** is equivalent to **(G1)**, and is introduced for better presentation, e.g., it connects to **(G2)** and **(M1)** through injectivity.

**(M1)** and **(D1)** are inherited from iVAE and are required for model (parameter) identifiability (identifying $\boldsymbol{f}_t$ up to affine mapping), which does not imply CATE identification in general. Arguably here the most important is that the mapping $\boldsymbol{f}_t$ from latent $Z$ to outcome $Y$ is injective, or else some information of $Z$ is in principle unrecoverable. These two conditions are not required by Proposition 2 which does not need model identifiability.

**(M2)**, together with overlapping PtSs, is important to address limited overlap of $X$ and can be seen as a weak form of OOD generalization.

**(M3')** means 1) we need to know or learn the distribution of hidden noise e and 2) noiseless prior. This simplifies the proof of identification, but when implementing the VAE as an estimation method, both noises are learned.

**(D2)**, or in fact **(D2')**, strengthens the model identifiability to determine both $\boldsymbol{f}_0$ and $\boldsymbol{f}_1$ up to the *same* affine mapping, which replaces the balance of PS.

**(G2)** is required by Proposition 2 but not Theorem 1. It is no less important than **(G1')**, because the core intuition of our method is that **(G2)** should hold approximately. Sec. C.3 contains several detailed real-world examples on **(G2)**.

## C.2 Details and Explanations on Intact-VAE

Our goal is to build a model that can be learned by VAE from observational data to obtain a PGS, or more ideally bPGS, via the latent variable $Z$. That is, a generative prognostic model. Generative models are useful to solve the inverse problem of recovering PGSs.

With the above goal, the generative model of our VAE is built as (3). Conditioning on $X$ in the joint model $p(\boldsymbol{y}, \boldsymbol{z}|\boldsymbol{x}, t)$ reflects that our estimand is CATE given $X$. Modeling the score by a conditional distribution rather than a deterministic function is more flexible.

The ELBO of our model can be derived from standard variational lower bound as following:

$$
\begin{aligned}
\log p(\boldsymbol{y}|\boldsymbol{x}, t) &\geq \log p(\boldsymbol{y}|\boldsymbol{x}, t) - D_{\mathrm{KL}}(q(\boldsymbol{z}|\boldsymbol{x}, \boldsymbol{y}, t)\|p(\boldsymbol{z}|\boldsymbol{x}, \boldsymbol{y}, t)) \\
&= \mathbb{E}_{\boldsymbol{z}\sim q} \log p(\boldsymbol{y}|\boldsymbol{z}, t) - D_{\mathrm{KL}}(q(\boldsymbol{z}|\boldsymbol{x}, \boldsymbol{y}, t)\|p(\boldsymbol{z}|\boldsymbol{x}, t)).
\end{aligned}
\tag{36}
$$

We naturally have an identifiable conditional VAE (CVAE), as the name suggests. Note that (3) has a similar factorization with the generative model of iVAE (Khemakhem et al., 2020a), that is $p(\boldsymbol{y}, \boldsymbol{z}|\boldsymbol{x}) = p(\boldsymbol{y}|\boldsymbol{z})p(\boldsymbol{z}|\boldsymbol{x})$; the first factor does not depend on $X$. Further, since we have the conditioning on $T$ in both the factors of (3), our VAE architecture is a combination of iVAE and CVAE (Sohn et al., 2015; Kingma et al., 2014), with $T$ as the conditioning variable. See Figure 1 for the comparison in terms of graphical models. The core idea of iVAE is reflected in our model identifiability (see Lemma 1).

Please do not confuse the DGP **(G1)** and the generative model (3) of Intact-VAE. The former is the causal model, but the latter is not (at least before we show the TE identifications in Sec. 3.2). In our case, the generative model is built as a way to learn the scores through the correspondence to (2).

In particular, note that conditionally balanced representation $Z \perp\!\!\!\perp T|X$ is possible under the generative model. This requires a violation of *causal faithfulness*, so that there are other conditional independence relations, which are not generally implied by the graphical model. Our method, based on iVAE, which achieves ICA, performs nonlinear ICA to recover the scores. In fact, ICA procedures often violate causal faithfulness, because it requires finding causes from effects. Also, the violation of causal faithfulness is not caused by the generative model (which is shown in Figure 1), because the representation is learned by the encoder, and $Z \perp\!\!\!\perp T|X$ is enforced by $\beta$.

## C.3 Discussions and examples of **(G2)**

We focus on univariate outcome on $\mathbb{R}$ which is the most practical case and the intuitions apply to more general types of outcomes. Then, $\boldsymbol{i}$, the mapping between $\mu_0$ and $\mu_1$, is monotone, i.e, either increasing or decreasing. The increasing $\boldsymbol{i}$ means, if a change of the value of $X$ increases (decreases) the outcome in the treatment group, then it is also the case for the controlled group. This is often true because the treatment does *not* change the mechanism how the covariates affect the outcome, under the principle of "independence of causal mechanisms (ICM)" (Janzing & Scholkopf, 2010). The decreasing $\boldsymbol{i}$ corresponds to another common interpretation when ICM does not hold. Now, the treatment does change the way covariates affect $Y$, but in a *global* manner: it acts like a "switch" on the mechanism: the same change of $X$ always has *opposite* effects on the two treatment groups.

We support the above reasoning by real world examples. First we give two examples where $\mu_0$ and $\mu_1$ are both monotone increasing. This, and also that both $\mu_t$ are monotone decreasing, are natural and sufficient conditions for increasing $\boldsymbol{i}$, though not necessary. The first example is form Health. (Starling et al., 2019) mentions that gestational age (length of pregnancy) has a monotone increasing effect on babies' birth weight, regardless of many other covariates. Thus, if we intervene on one of the other binary covariates (say, t = receive healthcare program or not), both $\mu_t$ should be monotone increasing in gestational age. The next example is from economics. (Gan & Li, 2016) shows that job-matching probability is monotone increasing in market size. Then, we can imagine that, with t = receive training in job finding or not, the monotonicity is not changed. Intuitively, the examples corresponds to two common scenarios: the causal effects are accumulated though time (the first example), or the link between a covariate and the outcome is direct and/or strong (the second example).

Examples for decreasing $\boldsymbol{i}$ are rarer and the following is a bit deliberate. This example is also about babies' birth weight as the outcome. (Abrevaya et al., 2015) shows that, with t = mother smokes

or not and $X$ = mother's age, the CATE $\tau(\boldsymbol{x})$ is monotone decreasing for $20 < \boldsymbol{x} < 26$ (smoking decreases birth weight, and the absolute causal effect is larger for older mother). On the other hand, it is shown that birth weight slightly increases (by about 100g) in the same age range in a surveyed population (Wang et al., 2020). Thus, it is convince that, smoking changes the the tendency of birth weight w.r.t mother's age from increasing to decreasing, and gives the large decreasing of birth weight (by about 300g) as its causal effect. This could be understood: the negative effects of smoking on mother's heath and in turn on birth weight are accumulated during the many years of smoking.

## C.4 COMPLEMENTARITY BETWEEN THE TWO IDENTIFICATIONS

We examine the complementarity between the two identifications more closely. The conditions **(M3) / (M3')** and **(G2) / (D2')** form two pairs, and are complementary inside each pair. The first pair matches model and truth, while the second pair restricts the discrepancy between the treatment groups. In Theorem 1, **(G2)** ($\mathfrak{p}_0 = \mathfrak{p}_1$) is replaced by **(D2')** which instead makes $\mathcal{A}_0 = \mathcal{A}_1 := \mathcal{A}$ in (5). And **(D2')** is easily satisfied with high-dimensional $X$, even if the possible values of $\boldsymbol{C}, \boldsymbol{d}$ are restricted to $\boldsymbol{C} = c\boldsymbol{I}$ and $\boldsymbol{d} = \boldsymbol{0}$ (see below). On the other hand, $p_\epsilon = p_\mathbf{e}$ in **(M3')** is impractical, but it ensures that $p_{\boldsymbol{\theta}}(\boldsymbol{y}|\boldsymbol{x}, t) = p(\boldsymbol{y}|\boldsymbol{x}, t)$ so that (5) can be used. In Sec. 4.1, we consider practical estimation method and introduce the *regularization* that encourages learning a PGS similar to bPGS so that $p_\epsilon = p_\mathbf{e}$ can be relaxed.

## C.5 IDEAS AND CONNECTIONS BEHIND THE ELBO (7)

**Bayesian approach is favorable** to express the prior belief that bPGSs exist and the preference for them, and to still have reasonable posterior estimation when the belief fails and learning general PGS is necessary. This is the causal importance of VAE as an estimation method for us. By the unconditional but still flexible $\boldsymbol{\Lambda}$, and also the identifications, the ELBO encourages the recovery of an approximate bPGS as the posterior, which still learns the dependence on $T$ if necessary. Moreover, $\beta$ expresses our additional knowledge (or, inductive bias) about whether or not there exist approximate bPGSs (e.g., from domain expertise).

In fact, $\beta$ connects our VAE to $\beta$-VAE (Higgins et al., 2017), which is closely related to noise and variance control (Doersch, 2016, Sec. 2.4)(Mathieu et al., 2019).

**Considerations on noise modeling.** In Theorem 1, with large and mismatched *noises* (then **(M3')** is easily violated), the identification of outcome model $\boldsymbol{f}_t = \boldsymbol{j}_t \circ \boldsymbol{v}^{-1}$ would fail, and, in turn, the prior would learn confounding bias, by confusing the causal effect of $T$ on $\mathfrak{p}_T$ and the correlation between $T$ and $X$. This is another reason to prefer $\boldsymbol{\lambda}_0 = \boldsymbol{\lambda}_1$, besides balancing. On the other hand, the posterior conditioning on $Y$ provides information of noise $\mathbf{e}$, and it is shown in (Bonhomme & Weidner, 2021) that posterior effect estimation has *minimum worst-case error* under model misspecification (of the noise and prior, in our case).

Under large $\mathbf{e}$, a relatively small $\beta$ implicitly encourages $\boldsymbol{g}$ *smaller* than the scale of $\mathbf{e}$, through stressing the third term in ELBO (7). And the the model as a whole would still learn $p(\boldsymbol{y}|\boldsymbol{x}, t)$ well, because the uncertainty of $\mathbf{e}$ can be moved to and modeled by the prior. This is why $\boldsymbol{k}$ is *not* set to zero because learnable prior noise (variance) allows us to implicitly control $\boldsymbol{g}$ via $\beta$. Intuitively, smaller $\boldsymbol{g}$ strengthens the correlation between $Y$ and $Z$ in our model, and this naturally reflects that posterior conditioning on $Y$ is more important under larger $\mathbf{e}$. Hopefully, precise learning of outcome noise **(M3')** is not required, as in Proposition 2.

Now, it is clear that $\beta$ naturally controls at the same time noise scale and balancing. And the regularization can also be understood as an interpolation between Proposition 2 and Theorem 1: relying on bPGS, or on model identifiability; learning loosely, or precisely, the outcome regression. When the noise scale is different from truth, there would be error due to imperfect recovery of $\boldsymbol{j}$. Sec. 4.2 shows that this error and balancing form a trade-off, which is adjusted by $\beta$.

**Importance of balancing from misspecification view.** If we must learn an unapproximate bPGS, we have larger misspecification under a balanced prior and rely more on $Y$ in the posterior. Both are bad because it is shown in (Bonhomme & Weidner, 2021) that posterior only helps under bounded (small) misspecification, and posterior estimator has higher variance than prior estimator (see below

for an extreme case). Again, we want a regularizer to encourage learning of bPGS, so that we can explore the *middle ground*: relatively low-dimensional $\mathfrak{p}$, or relatively small $\mathbf{e}$.

**Example.** Assume the true outcome noise is (near) zero. By setting $\epsilon \to \mathbf{0}$ in our model, the posterior $p_{\boldsymbol{\theta}}(\boldsymbol{z}|\boldsymbol{x}, \boldsymbol{y}, t) = p_{\boldsymbol{\theta}}(\boldsymbol{y}, \boldsymbol{z}|\boldsymbol{x}, t)/p_{\boldsymbol{\theta}}(\boldsymbol{y}|\boldsymbol{x}, t)$ degenerates to $\boldsymbol{f}_T^{-1}(Y) = \boldsymbol{f}_T^{-1}(\boldsymbol{j}_T(\mathfrak{p}_T)) = \boldsymbol{v}^{-1}(\mathfrak{p}_T)$, a *factual* PGS. However, $\boldsymbol{f}_{1-T}^{-1}(Y) = \boldsymbol{f}_{1-T}^{-1}(\boldsymbol{j}_T(\mathfrak{p}_T)) = \boldsymbol{v}^{-1}(\boldsymbol{j}_{1-T}^{-1} \circ \boldsymbol{j}_T(\mathfrak{p}_T)) \neq \boldsymbol{v}^{-1}(\mathfrak{p}_{1-T})$, *the score recovered by posterior does not work for counterfactual assignment*! The problem is, unlike $X$, the outcome $Y = Y(T)$ is affected by $T$, and, the degenerated posterior disregards the information of $X$ from the prior and depends exclusively on factual $(Y, T)$.

## C.6 CONSISTENCY OF VAE AND PRIOR ESTIMATION

The following is a refined version of Theorem 4 in Khemakhem et al. (2020a). The result is proved by assuming: i) our VAE is flexible enough to ensure the ELBO is tight (equals to the true log likelihood) for some parameters; ii) the optimization algorithm can achieve the *global* maximum of ELBO (again equals to the log likelihood).

**Proposition 6** (Consistency of Intact-VAE). *Given model* (3)&(6)*, and let $p^*(\boldsymbol{x}, \boldsymbol{y}, t)$ be the true observational distribution, assume*

> i) *there exists $(\bar{\boldsymbol{\theta}}, \bar{\boldsymbol{\phi}})$ such that $p_{\bar{\boldsymbol{\theta}}}(\boldsymbol{y}|\boldsymbol{x}, t) = p^*(\boldsymbol{y}|\boldsymbol{x}, t)$ and $p_{\bar{\boldsymbol{\theta}}}(\boldsymbol{z}|\boldsymbol{x}, \boldsymbol{y}, t) = q_{\bar{\boldsymbol{\phi}}}(\boldsymbol{z}|\boldsymbol{x}, \boldsymbol{y}, t)$;*
> ii) *the ELBO $\mathbb{E}_{\mathcal{D} \sim p^*}(\mathcal{L}(\boldsymbol{x}, \boldsymbol{y}, t; \boldsymbol{\theta}, \boldsymbol{\phi}))$ (4) can be optimized to its global maximum at $(\boldsymbol{\theta}', \boldsymbol{\phi}')$;*

*Then, in the limit of infinite data, $p_{\boldsymbol{\theta}'}(\boldsymbol{y}|\boldsymbol{x}, t) = p^*(\boldsymbol{y}|\boldsymbol{x}, t)$ and $p_{\boldsymbol{\theta}'}(\boldsymbol{z}|\boldsymbol{x}, \boldsymbol{y}, t) = q_{\boldsymbol{\phi}'}(\boldsymbol{z}|\boldsymbol{x}, \boldsymbol{y}, t)$.*

*Proof.* From i), we have $\mathcal{L}(\boldsymbol{x}, \boldsymbol{y}, t; \bar{\boldsymbol{\theta}}, \bar{\boldsymbol{\phi}}) = \log p^*(\boldsymbol{y}|\boldsymbol{x}, t)$. But we know $\mathcal{L}$ is upper-bounded by $\log p^*(\boldsymbol{y}|\boldsymbol{x}, t)$. So, $\mathbb{E}_{\mathcal{D} \sim p^*}(\log p^*(\boldsymbol{y}|\boldsymbol{x}, t))$ should be the global maximum of the ELBO (even if the data is finite).

Moreover, note that, for any $(\boldsymbol{\theta}, \boldsymbol{\phi})$, we have $D_{\mathrm{KL}}(p_{\boldsymbol{\theta}}(\boldsymbol{z}|\boldsymbol{x}, \boldsymbol{y}, t) \| q_{\boldsymbol{\phi}}(\boldsymbol{z}|\boldsymbol{x}, \boldsymbol{y}, t) \geq 0$ and, in the limit of infinite data, $\mathbb{E}_{\mathcal{D} \sim p^*}(\log p_{\boldsymbol{\theta}}(\boldsymbol{y}|\boldsymbol{x}, t)) \leq \mathbb{E}_{\mathcal{D} \sim p^*}(\log p^*(\boldsymbol{y}|\boldsymbol{x}, t))$. Thus, the global maximum of ELBO is achieved *only* when $p_{\boldsymbol{\theta}}(\boldsymbol{y}|\boldsymbol{x}, t) = p^*(\boldsymbol{y}|\boldsymbol{x}, t)$ and $p_{\boldsymbol{\theta}}(\boldsymbol{z}|\boldsymbol{x}, \boldsymbol{y}, t) = q_{\boldsymbol{\phi}}(\boldsymbol{z}|\boldsymbol{x}, \boldsymbol{y}, t)$. $\square$

Consistent prior estimation of CATE follows directly from the identifications. The following is a corollary of Theorem 1.

**Corollary 1.** *Under the conditions of Theorem 1, further require the consistency of Intact-VAE. Then, in the limit of infinite data, we have $\mu_t(X) = \boldsymbol{f}_t(\boldsymbol{h}_t(X))$ where $\boldsymbol{f}, \boldsymbol{h}$ are the optimal parameters learned by the VAE.*

## C.7 PRE/POST-TREATMENT PREDICTION

Sampling posterior requires *post-treatment* observation $(\boldsymbol{y}, t)$. Often, it is desirable that we can also have *pre-treatment* prediction for a new subject, with only the observation of its covariate $X = \boldsymbol{x}$. To this end, we use the prior as a pre-treatment predictor for $Z$: replace $q_{\boldsymbol{\phi}}$ with $p_{\boldsymbol{\Lambda}}$ in (8) and get rid of the outer average taken on $\mathcal{D}$; all the others remain the same. We also have sensible pre-treatment prediction even without true low-dimensional PSs, because $p_{\boldsymbol{\Lambda}}$ gives the best balanced approximation of the target PGS. The results of pre-treatment prediction are given in the experimental section E.

# D MORE ON RELATED WORK

## D.1 CFR AND CEVAE

CFR and CEVAE are well-known machine learning methods for CATE estimation. Here we make detailed comparisons to them.

### D.1.1 COMPARISONS WITH CFR

Our method is related to CFR in two ways. Theoretically, our bounds in Sec. 4.2 resemble those in Shalit et al. (2017). But we bound CATE error, while CFR bounds PEHE; thus, our bounds give

conditional balancing while CFR only has unconditional balancing. See Sec. D.3 for more on the bounds. Conceptually, CFR is loosely related to our method because it also learns a representation as an outcome predictor, as mentioned in the follow-up Johansson et al. (2020). However, CFR does not have a generative model, so their representation is not formally related to PGSs. Moreover, CFR does not account the outcome noise, while the uncertainty due to the noise is accounted by our VAE.

### D.1.2 Comparisons with and criticisms of CEVAE

**Motivation**   CEVAE is motivated by exploiting proxy variables, and its intuition is that the hidden confounder $U$ can be recovered by VAE from proxy variables.

Our method is motivated by prognostic scores (Hansen, 2008), and our model is directly based on equations (2) which identifies CATE. There is no need to recover the hidden confounder in our framework.

**Architecture**   Our model is naturally based on (2), particularly the independence properties of PGS. And as a consequence, our VAE architecture is a natural combination of iVAE and CVAE (see Figure 1). Our ELBO (4) is derived by the standard variational lower bound.

On the other hand, the architecture of CEVAE is more ad hoc and complex. Its decoder follows the graphical model of descendant proxy mentioned above, but adds an ad hoc component to mimic TARnet (Shalit et al., 2017): it uses separated NNs for the two potential outcomes. We tried this idea on the IHDP dataset, and, as we show in Sec. 5.2, it has basically no merits for our method, because we have a principled way for balancing.

The encoder of CEVAE is even more complex. To have post-treatment estimation, $q(T|X)$ and $q(Y|X,T)$ are added into the encoder. As a result, the ELBO of CEVAE has two additional likelihood terms corresponding to the two distributions. However, in our Intact-VAE, post-treatment estimation is given naturally by our standard encoder, thanks to the correspondence between our model and (2).

**Justification**   We have given the identifications and bounds of our method in this paper. Moreover, we carefully distinguish assumptions on the DGP and assumptions on our model, and identify the assumptions that are important for causality. There are few theoretical justifications for CEVAE. Their Theorem 1 directly assumes the joint distribution $p(\boldsymbol{x}, \boldsymbol{y}, t, \boldsymbol{u})$ including hidden confounder $U$ is recovered, then identification is trivial by using the standard adjustment equation.

However, the challenge is exactly that the confounder is hidden, unobserved. Many years of work have been done in causal inference to derive conditions under which hidden confounder can be (partially) recovered (Greenland, 1980; Kuroki & Pearl, 2014; Miao et al., 2018). In particular, Miao et al. (2018) gives the most recent identification result for proxy setting, which requires very specific two proxies structure, and other completeness assumptions on distributions. Thus, it is unreasonable to believe that VAE, with simple descendant proxies, can recover the hidden confounder. Indeed, Rissanen & Marttinen (2021) recently give evidence that the method often fails.

Moreover, the identifiability of VAE itself is a challenging problem. As mentioned in Introduction, Khemakhem et al. (2020a) is the first identifiability result for VAE, but it only identifies an equivalence class, not a unique representation function. Thus, it is also unconvincing that VAE can learn a unique latent distribution, without certain assumptions. As we show in Sec. 5.1, for relatively simple synthetic datasets, CEVAE can not robustly recover the hidden confounder, even only up to transformation, while our method can (though, again, this is not needed for our method).

### D.2 Injectivity, invertibility, monotonicity, and overlap

Let us note that *any injective mapping defines an invertible mapping*, by restrict the domain of the inverse function to the range of the injective mapping. Also note that injectivity is weaker than monotonicity; a monotone mapping can be defined by an injective and *order-preserving* mapping between ordered sets. Particularly, *an injective and continuous mapping on $\mathbb{R}$ is monotone*, and many works in econometrics give examples of this case.

Many classical and recent works (with many real world applications, see C.1) in econometrics are based on monotonicity. Particularly, there is a long line of work based on *monotonicity of treatment* (Huber & Wüthrich, 2018). More related to our method is another line of work based on *monotonicity of outcome*, see (Chernozhukov & Hansen, 2013) and references therein for early results. Some recent works apply monotonicity of outcome to nonparametric IV regression (NPIV) (Freyberger & Horowitz, 2015; Li et al., 2017; Chetverikov & Wilhelm, 2017), where the structural equation of the outcome is assumed to be $Y = f(T) + \epsilon$, and $f$ is monotone and $T$ (the treatment) is often continuous. Particularly, (Chetverikov & Wilhelm, 2017) combines monotonicity of both treatment and outcome, and (Freyberger & Horowitz, 2015) considers *discrete* treatment (note continuity or differentiability is not necessary for monotonicity). NPIV with monotone $f$ is closely related to our method, but the difference is that $T$ is replaced by a PGS in our method, and the PGS is recovered from observables. Finally, as we mentioned in Sec. 3.2, monotonicity is a kind of shape restriction which also includes, e.g., concavity and symmetry and attracts recent interests (Chetverikov et al., 2018). However, most of NPIV works focus on identifying $f$ but not directly on TEs, and we do not know any works that use monotonicity to address limited overlap.

Recently in machine learning, (Johansson et al., 2019; Zhang et al., 2020b; Johansson et al., 2020) note the relationship between invertibility and overlap. As mentioned, (Johansson et al., 2020) gives bounds without overlap, but the relationship between invertibility and overlap is not explicit in their theory. (Johansson et al., 2019) explicitly discuss overlap and invertibility, but does not focus on TEs. (Zhang et al., 2020b) assumes overlap so that identification is given, and then focuses on learning overlapping representation that preserves the overlapping the covariate. However, it does not relate invertibility and overlap, but uses invertible representation function to *preserve exchangeability given the covariate*, and linear outcome regression to simply the model. Related, our identifications required **(M2)**, of which linearity of PGS and representation function is a sufficient condition, and our outcome model is injective, to *preserve the exchangeability given the PGS*. Thus, our method works under more general setting, and arguably under weaker conditions.

### D.3 ADDITIONAL NOTES ON NOVELTIES OF THE BOUNDS IN SEC. 4.2

We give details and additional points regarding the novelties. Lu et al. (2020) also use a VAE and derive bounds most related to ours. Still, our method strengthens Lu et al. (2020), in a simpler and principled way: we distinguish true score and latent $Z$ and show that identification is the link; considering both prior and posterior, we show the symmetric nature of the balancing term and relate it to our KL term in (7), without ad hoc regularization; moreover, we consider outcome noise modeling which is a strength of VAE and relate it to hyperparameter $\beta$. Particularly, in (Lu et al., 2020), latent variable $Z$ is confused with the true representation ($\mathfrak{p}_t$ up to invertible mapping in our case). *Without* identification, the method in fact has unbounded error. Note that Shalit et al. (2017) do not consider connection to identification and noise modeling as well. The error between $\hat{\tau}_f$ and $\tau_j$, which we bound, is due to the unknown outcome noise that is not accounted by our Theorem 1; thus, the theory in Sec. 4.2 is complementary to that in Sec. 3.2. Finally, $\beta$ is a trade-off between the conditional balance of learned PGS (affected by $\boldsymbol{f}_t$), and precision/effective sample size of outcome regression—and can be seen as the probabilistic counterpart of Tarr & Imai (2021) and Kallus et al. (2018).

## E DETAILS AND ADDITIONS OF EXPERIMENTS

We evaluate the post-treatment performance on training and validation set jointly (This is non-trivial. Recall the fundamental problem of causal inference). The treatment and (factual) outcome should not be observed for pre-treatment predictions, so we report them on a testing set. See also Sec. C.7 the pre/post-treatment distinction.

## E.1 Synthetic data

We detail how the random parameters in the DGPs are sampled. $\mu_i$ and $\sigma_i$ are uniformly sampled in range $(-0.2, 0.2)$ and $(0, 0.2)$, respectively. The weights of linear functions $\boldsymbol{h}, \boldsymbol{k}, l$ are sampled from standard normal distributions. The NNs $f_0, f_1$ use leaky ReLU activation with $\alpha = 0.5$ and are of 3 to 8 layers randomly, and the weights of each layer are sampled from $(-1.1, -0.9)$. To have a large but still reasonable outcome variance, the output of $f_t$ is divided by $C_t := \mathrm{Var}_{\{\mathcal{D}|T=t\}}(f_t(Z))$. When generating DGPs with dependent noise, the variance parameter $g_t$ for the outcome is generated by adding a softplus layer after respective $f_t$, and then normalized to range $(0, 2)$.

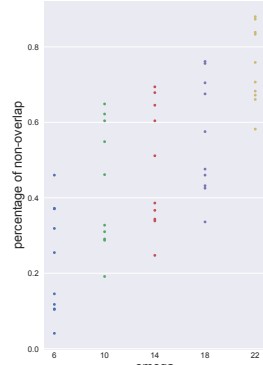

Figure 4: Degree of limited overlap w.r.t $\omega$.

We use the original implementation of CFR[3]. Very possibly due to bugs in implementation, the CFR version using Wasserstein distance has error of TensorFlow type mismatch on our synthetic dataset, and the CFR version using MMD diverges with very large loss value on one or two of the 10 random DGPs. We use MMD version, and, when the divergence of training happens, report the results from trained models before divergence, which still give reasonable results. We search the balancing parameter alpha in [0.16, 0.32, 0.64, 0.8, 1.28], and fix other hyperparameters as they were in the default config file.

We characterize the degree of limited overlap by examining the percentage of observed values $\boldsymbol{x}$ that give probability less than 0.001 for one of $p(t|\boldsymbol{x})$. The threshold is chosen so that all sample points near those values $\boldsymbol{x}$ almost certainly belong to a single group since we have 500 sample point in total. If we regard a DGP as very limited-overlapping when the above percentage is larger than 50%, then, as shown in Figure 4, non (all) of the 10 DGPs are very limited-overlapping with $\omega = 6$ ($\omega = 22$).

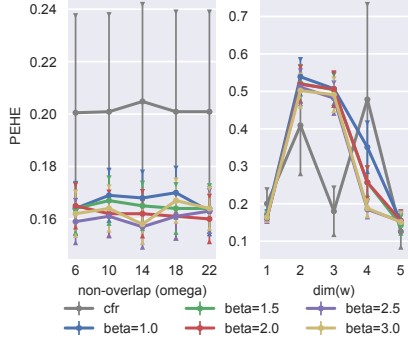

Figure 5: $\sqrt{\epsilon_{pehe}}$ on synthetic dataset, with $g_t(W) = 1$ in DGPs, and $\dim(Z) = 200$ in our model. Error bar on 10 random DGPs.

For diversity of the datasets, we set $g_t(W) = 1$ in DGPs in Appendix. Figure 5 shows, with $\dim(Z) = 200$, our method works better than CFR under $\dim(W) = 1$ and as well as CFR under $\dim(W) > 1$. As mentioned in Conclusion, this indicates that the theoretical requirement of injective $\boldsymbol{f}_t$ in our model might be relaxed. Interestingly, larger $\beta$ seems to give better results here, this is understandable because $\beta$ controls the trade-off between fitting and balancing, and the fitting capacity of our decoder is much increased with $\dim(Z) = 200$. Note that the above observations on $\dim(Z)$ are not caused by fixing $g_t(W) = 1$ (compare Figure 5 with Figure 6 below).

Figure 6 shows the importance of noise modeling. Compared to Figure 2 in the main text, where $g_t(W)$ in DGPs is not fixed, our method works worse here, particularly for large $\beta$, because now noise modeling ($\boldsymbol{g}, \boldsymbol{k}$ in the ELBO) only adds unnecessary complexity. The changes of performance w.r.t different $\omega$ should be unrelated to overlap levels, but to the complexity of random DGPs; compare to Figure 5, with larger NNs in our VAE, the changes become much insignificant. The drop of error for $\dim(W) > 3$ is due to the randomness of $f$ in (36). In Sec. 2.2, we saw that the 2-dimensional bPGS $\mathfrak{p} := (\mu_0(X), \mu_1(X))$ always exists under additive noise models. Thus, when $\dim(W) > 2$, our method tries to recover that $\mathfrak{p}$, and generally performs not worse than under $\dim(W) = 2$, but still not better than under $\dim(W) = 1$.

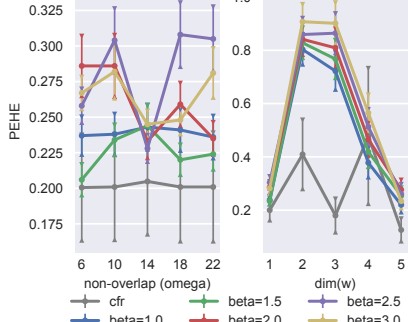

Figure 6: $\sqrt{\epsilon_{pehe}}$ on synthetic dataset, with $g_t(W) = 1$ in DGPs. Error bar on 10 random DGPs.

---

[3]https://github.com/clinicalml/cfrnet

Figure 7 shows results of ATE estimation. Notably, CFR drops performance w.r.t degree of limited overlap. Our method does not show this tendency except for very large $\beta$ ($\beta = 3$). This might be another evidence that CFR and its unconditional balancing overfit to PEHE (see Sec. 5.2). Also note that, under $\dim(W) = 1$, $\beta = 3$ gives the best results for ATE although it does not work well for PEHE, and we do not know if this generalizes to the conclusion that large $\beta$ gives better ATE estimation under the existence of bPGS, but leave this for future investigation.

Figure 8 shows results of pre-treatment prediction. In left panel, both our method and CFR perform only slightly worse than post-treatment. This is reasonable because here we have bPGS $W$ with $\dim(W) = 1$, there is no need to learn PGS. In the right panel, we also do not see significant drop of performance compared to post-treatment. This might be due to the hardness of learning approximate bPGS in this dataset, and posterior estimation does not give much improvements.

You can find more plots for latent recovery at the end of the paper.

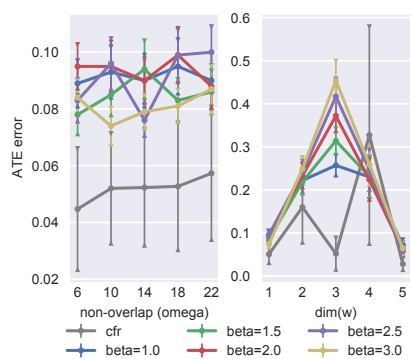

Figure 7: $\epsilon_{ate}$ on synthetic dataset, with $g_t(W) = 1$ in DGPs. Error bar on 10 random DGPs.

## E.2 IHDP

IHDP is based on an RCT where each data point represents a baby with 25 features (6 continuous, 19 binary) about their birth and mothers. `Race` is introduced as a confounder by artificially removing all treated children with nonwhite mothers. There are 747 subjects left in the dataset. The outcome is synthesized by taking the covariates (features excluding `Race`) as input, hence *unconfoundedness* holds given the covariates. Following previous work, we split the dataset by 63:27:10 for training, validation, and testing. Note, there is no ethical concerns here, because the treatment assignment mechanism is artificial by processing the data. Also our results are only quantitative and we make no ethical conclusions.

The generating process is as following (Hill, 2011, Sec. 4.1).

$$Y(0) \sim \mathcal{N}(e^{\boldsymbol{a}^T(X+\boldsymbol{b})}, 1), \quad Y(1) \sim \mathcal{N}(\boldsymbol{a}^T X - c, 1), \tag{37}$$

where $\boldsymbol{a}$ is a random coefficient, $\boldsymbol{b}$ is a constant bias with all elements equal to $0.5$, and $c$ is a random parameter adjusting degree of overlapping between the treatment groups. As we can see, $\boldsymbol{a}^T X$ is a true bPGS. As mentioned in the main text, the bPGS might be discrete. Thus, this experiment also shows the importance of VAE, even if an apparent bPGS exists. Under *discrete* PSs, training an regression based on Proposition 2 is hard, but our VAE works well.

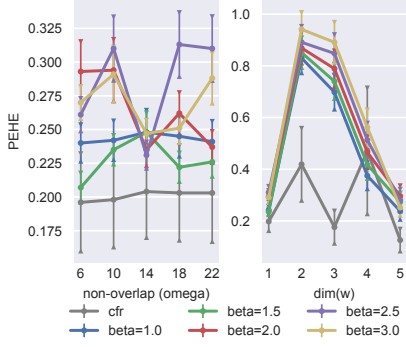

Figure 8: *Pre-treatment* $\sqrt{\epsilon_{pehe}}$ on synthetic dataset. Error bar on 10 random DGPs.

The two added components in the modified version of our method are as following. First, we build the two outcome functions $\boldsymbol{f}_t(Z), t = 0, 1$ in our learning model (3), using two separate NNs. Second, we add to our ELBO (4) a regularization term, which is the Wasserstein distance (Cuturi, 2013) between $\mathbb{E}_{\mathcal{D} \sim p(X|T=t)} p_{\boldsymbol{\Lambda}}(Z|X), t \in \{0, 1\}$. As shown in Table 2, best unconditional balancing parameter is 0.1. Larger parameters gives much worse PEHE and does not improve ATE estimation. Smaller parameters are more reasonable but still do not improve the results. The overall tendency is clear. Compared to ours, CFR with its unconditional balancing does not improve ATE estimation, it may improve PEHE results with fine tuned parameter, but possibly at the price of worse ATE estimation.

Table 3 shows pre-treatment results, All methods gives reasonable results.

Table 2: Performance of modified version with different unconditional balancing parameter, the values of which are shown after "Mod.".

| Method | Ours | Mod. 1 | Mod. 0.2 | Mod. 0.1 | Mod. 0.05 | Mod. 0.01 | CFR |
|---|---|---|---|---|---|---|---|
| $\epsilon_{ate}$ | $.177_{\pm.007}$ | $.196_{\pm.008}$ | $.177_{\pm.007}$ | $.167_{\pm.005}$ | $.177_{\pm.006}$ | $.179_{\pm.006}$ | $.25_{\pm.01}$ |
| $\sqrt{\epsilon_{pehe}}$ | $.843_{\pm.030}$ | $1.979_{\pm.082}$ | $1.116_{\pm.046}$ | $.777_{\pm.026}$ | $.894_{\pm.039}$ | $.841_{\pm.029}$ | $.71_{\pm.02}$ |

Table 3: *Pre-treatment* Errors on IHDP over 1000 random DGPs. We report results with $\dim(Z) = 10$. **Bold** indicates method(s) which is *significantly* better. The results are taken from Shalit et al. (2017), except GANITE (Yoon et al., 2018) and CEVAE (Louizos et al., 2017).

| Method | TMLE | BNN | CFR | CF | CEVAE | GANITE | Ours |
|---|---|---|---|---|---|---|---|
| pre-$\epsilon_{ate}$ | NA | $.42_{\pm.03}$ | $.27_{\pm.01}$ | $.40_{\pm.03}$ | $.46_{\pm.02}$ | $.49_{\pm.05}$ | $\mathbf{.211}_{\pm.011}$ |
| pre-$\sqrt{\epsilon_{pehe}}$ | NA | $2.1_{\pm.1}$ | $\mathbf{.76}_{\pm.02}$ | $3.8_{\pm.2}$ | $2.6_{\pm.1}$ | $2.4_{\pm.4}$ | $.946_{\pm.048}$ |

### E.3 POKEC SOCIAL NETWORK DATASET

This experiment shows our method is the best compared with the methods specialized for networked deconfounding, a challenging problem in its own right. Thus, our method has the potential to work under *unobserved confounding*, but we leave detailed experimental and theoretical investigation to future.

Pokec (Leskovec & Krevl, 2014) is a real world social network dataset. We experiment on a semi-synthetic dataset based on Pokec, which was introduced in (Veitch et al., 2019), and use exactly the same pre-processing and generating procedure. The pre-processed network has about 79,000 vertexes (users) connected by $1.3 \times 10^6$ undirected edges. The subset of users used here are restricted to three living districts which are within the same region. The network structure is expressed by binary adjacency matrix $\boldsymbol{G}$. Following (Veitch et al., 2019), we split the users into 10 folds, test on each fold and report the mean and std of pre-treatment ATE predictions. We further separate the rest of users (in the other 9 folds) by $6:3$, for training and validation.

Each user has 12 attributes, among which `district`, `age`, or `join date` is used as a confounder $U$ to build 3 different datasets, with remaining 11 attributes used as covariate $X$. Treatment $T$ and outcome $Y$ are synthesised as following:

$$T \sim \text{Bern}(g(U)), \quad Y = T + 10(g(U) - 0.5) + \epsilon, \tag{38}$$

where $\epsilon$ is standard normal. Note that `district` is of 3 categories; `age` and `join date` are also discretized into three bins. $g(U)$, which is a bPGS, maps these three categories and values to $\{0.15, 0.5, 0.85\}$.

$\beta$-Intact-VAE is expected to learn a bPGS from $\boldsymbol{G}, X$, if we can exploit the network structure effectively. Given the huge network structure, most users can practically be identified by their attributes and neighborhood structure, which means $U$ can be roughly seen as a deterministic function of $\boldsymbol{G}, X$. This idea is comparable to Assumptions 2 and 4 in (Veitch et al., 2019), which postulate directly that a balancing score can be learned in the limit of infinite large network. To extract information from the network structure, we use Graph Convolutional Network (GCN) (Kipf & Welling, 2017) in conditional prior and encoder of $\beta$-Intact-VAE. The implementation details are given at the end of this subsection.

Table 4 shows the results. The pre-treatment $\sqrt{\epsilon_{pehe}}$ for `Age`, `District`, and `Join date` confounders are 1.085, 0.686, and 0.699 respectively, practically the same as the ATE errors. Note that, Veitch et al. (2019) does not give individual-level prediction.

To extract information from the network structure, we use Graph Convolutional Network (GCN) (Kipf & Welling, 2017) in conditional prior and encoder of $\beta$-Intact-VAE. A difficulty is that, the network $\boldsymbol{G}$ and covariates $\boldsymbol{X}$ of *all* users are always needed by GCN, regardless of whether it is in training, validation, or testing phase. However, the separation can still make sense if we take care that the treatment and outcome are used only in the respective phase, e.g., $(y_m, t_m)$ of a testing user $m$ is only used in testing.

Table 4: Pre-treatment ATE on Pokec. Ground truth ATE is 1, as we can see in (38). "Unadjusted" estimates ATE by $\mathbb{E}_{\mathcal{D}}(y_1) - \mathbb{E}_{\mathcal{D}}(y_0)$. "Parametric" is a stochastic block model for networked data (Gopalan & Blei, 2013). "Embed-" denotes the best alternatives given by (Veitch et al., 2019). **Bold** indicates method(s) which is *significantly* better than all the others. We report results with 20-dimensional latent $Z$. The results of the other methods are taken from (Veitch et al., 2019).

|  | Age | District | Join Date |
|---|---|---|---|
| Unadjusted | $4.34 \pm 0.05$ | $4.51 \pm 0.05$ | $4.03 \pm 0.06$ |
| Parametric | $4.06 \pm 0.01$ | $3.22 \pm 0.01$ | $3.73 \pm 0.01$ |
| Embedding-Reg. | $2.77 \pm 0.35$ | $\mathbf{1.75 \pm 0.20}$ | $2.41 \pm 0.45$ |
| Embedding-IPW | $3.12 \pm 0.06$ | $\mathbf{1.66 \pm 0.07}$ | $3.10 \pm 0.07$ |
| Ours | $\mathbf{2.08 \pm 0.32}$ | $\mathbf{1.68 \pm 0.10}$ | $\mathbf{1.70 \pm 0.13}$ |

GCN takes the network matrix $G$ and the *whole* covariates matrix $X := (x_1^T, \ldots, x_M^T)^T$, where $M$ is user number, and outputs a representation matrix $R$, again for all users. During training, we *select* the rows in $R$ that correspond to users in training set. Then, treat this *training representation matrix* as if it is the covariates matrix for a non-networked dataset, that is, the downstream networks in conditional prior and encoder are the same as in the other two experiments, but take $(R_{m,:})^T$ where $x_m$ was expected as input. And we have respective selection operations for validation and testing. We can still train $\beta$-Intact-VAE including GCN by Adam, simply setting the gradients of non-seleted rows of $R$ to 0.

Note that GCN cannot be trained using mini-batch, instead, we perform batch gradient decent using full dataset for each iteration, with initial learning rate $10^{-2}$. We use dropout (Srivastava et al., 2014) with rate 0.1 to prevent overfitting.

### E.4 EMPIRICAL VALIDATION OF THE BOUNDS IN SEC. 4.2

Here we focus on the $D(X)$ term in Theorem 2 because it is directly related to conditional balance.

In the Figure attached at the end of the paper, the rows correspond to 3 overlap levels from strong to weak ($\omega = 6, 14, 22$ respectively). The first column shows the histograms of correlation coefficients between $D(X)$ and $\epsilon_f(X)$ on 100 random DGPs. The vertical bars in the histograms are 5, 25, 50, 75, 95 percentiles (the values are shown in the table below). The other 10 columns show the plots of distributions of $(D(X), \epsilon_f(X))$ for the first 10 DGPs. The correlation coefficient for each DGP is shown as `corrcoef=*` above each histogram. The plots are in log-log scale, because both $D$ and $\epsilon_f$ are single-sided, and most data points concentrate near $(0, 0)$, making the plots bad-looking.

We have two important observations from the histograms: 1) on the majority of DGPs, there are positive correlations between $D$ and $\epsilon_f$; 2) the positive correlation is stronger with weaker overlap (the portion of large correlation increases, and the mean `corrcoef` are 0.100, 0.110, and 0.121, respectively).

Thus, our bounds and conditional balance have significance. Not all DGPs have positive correlations, and this is reasonable because our bound (11) has three other terms which can obscure the relation between $D$ and $\epsilon_f$. The DGPs 1, 3, 6, 8, 10 show typical situations when there are positive correlations.

Table 5: Percentiles of correlation coefficients between $D(X)$ and $\epsilon_f(X)$ on 100 random DGPs.

| Percentile | 5 | 25 | 50 | 75 | 95 |
|---|---|---|---|---|---|
| $\omega = 6$ | -0.289 | -0.086 | 0.069 | 0.299 | 0.609 |
| $\omega = 14$ | -0.328 | -0.124 | 0.055 | 0.337 | 0.636 |
| $\omega = 22$ | -0.274 | -0.128 | 0.067 | 0.341 | 0.634 |

### E.5 ADDITIONAL PLOTS ON SYNTHETIC DATASETS

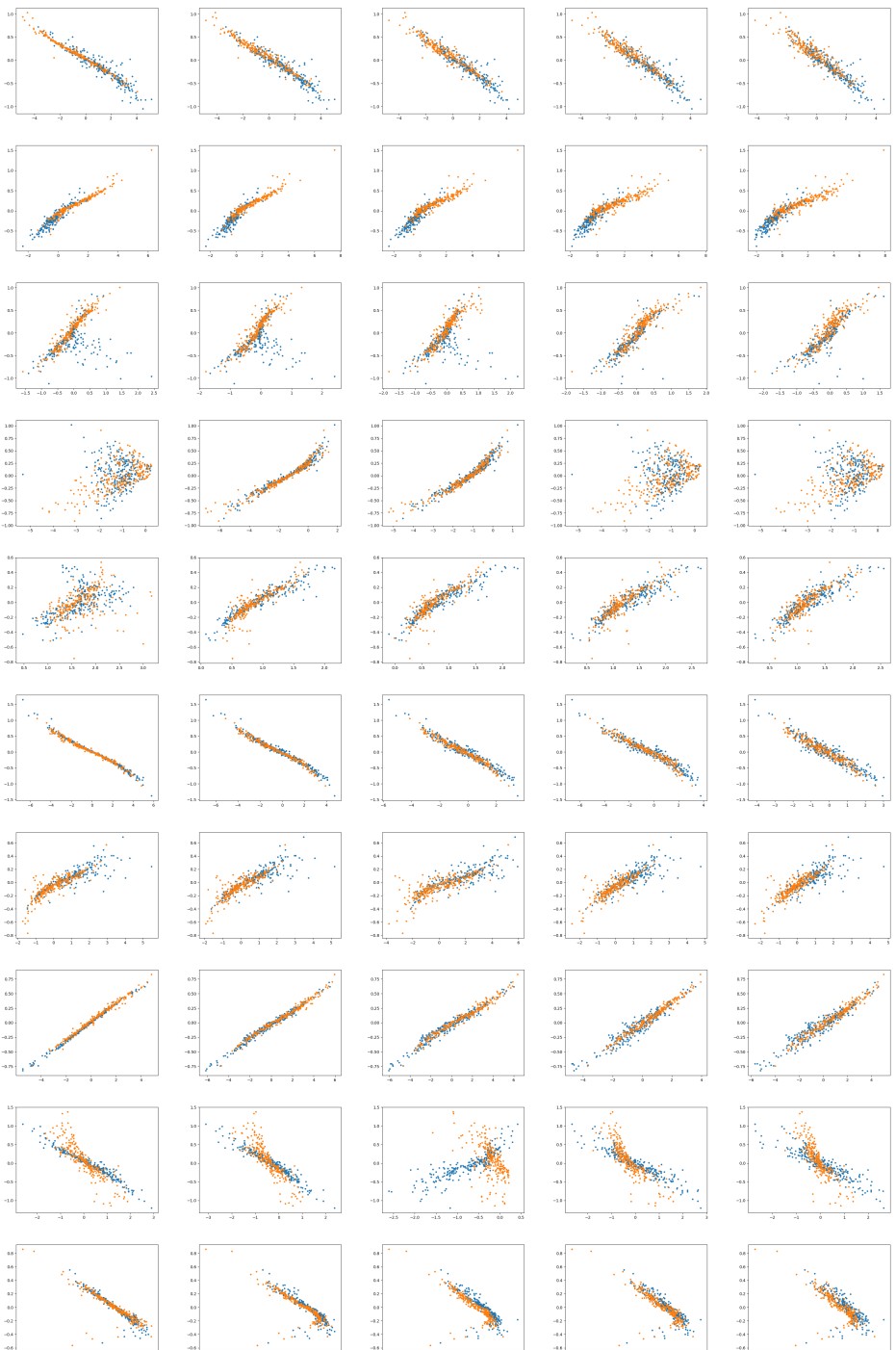

Figure 9: Plots of recovered-true latent. Rows: first 10 nonlinear random models, columns: outcome noise level.

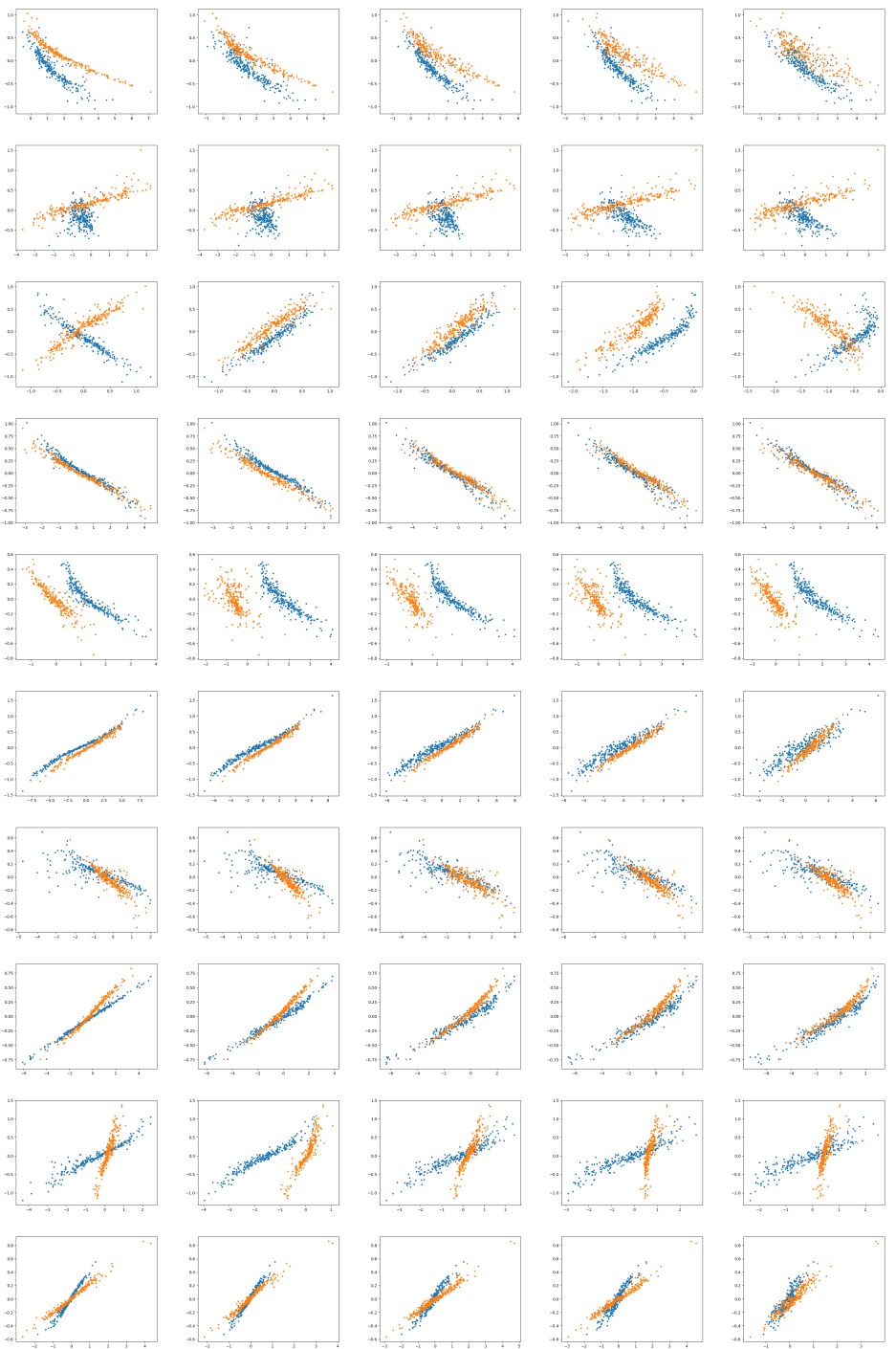

Figure 10: Plots of recovered-true latent. Conditional prior *depends* on $t$. Rows: first 10 nonlinear random models, columns: outcome noise level. Compare to the previous figure, we can see the transformations for $t = 0, 1$ are *not* the same, confirming the importance of balanced prior.

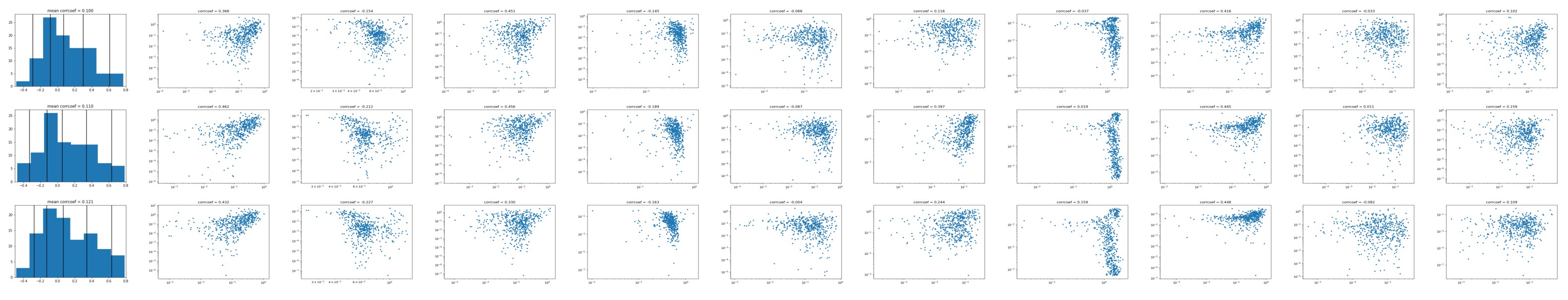