# OpenReview forum: "$\beta$-Intact-VAE: Identifying and Estimating Causal Effects under Limited Overlap"
_ICLR.cc/2022/Conference — ICLR 2022 Poster_

### Official Review · Reviewer_ENDL · 2021-11-01

**Correctness:** 4
**Technical Novelty And Significance:** 3
**Empirical Novelty And Significance:** 3
**Recommendation:** 6
**Confidence:** 3

**Main Review:**

Strengths:

1. Authors contribute a novel approach for limited overlapping of covariates. The setting of limited overlapping is interesting. The proposed approach combines the prognostic score and Intact-VAE, which are appropriate and novel under this setting. Although the \beta variational lower bound was proposed in the previous methods, it plays a role of controlling balance and it can be viewed as a novel application.

2. The experiment results are extensive. The authors follow some existing data generation processing or data and show that the balanced prior is important in practice compared to non-balanced ones and their models can obtain better performance.

3. The theoretical analyses are novel under the setting of this paper. These results follow those of iVAE and adapt to the new settings. The authors elaborate on some conditions to help readers to get them, which is great.

Weaknesses:

1. The paper is dense and difficult to follow. It introduces many concepts from different areas. It takes much time to understand and check the paper. I suggest authors add some plots to help readers to faster understand some definitions, e.g. overlapping. The paper involves and requires readers to have wide knowledge, so I think that a good presentation is difficult. Maybe moving some contents to the appendix and illustrating some concepts could be a choice. In addition, characters with the same style are used for different types of subjects, such as $\mathbb{R}$ for real numbers and $\mathbb{M}$ for a function. I strongly suggest using consistent notations.

2. Experiment results measured by pehe are worse than CFR. ATE and PEHE reflect different information. ATE focuses on the mean and PEHE reflects the stability because if there is a large CATE error, PEHE can be large (as mentioned on the page 9). Large CATE errors mean that the model performs unstably at some values. So, it is better to discuss the results more properly.


**Summary Of The Paper:**

The paper considers the setting of limited overlap of covariates and studies identifying and estimating the causal effects. This setting is difficult to deal with because it is impossible to estimate causal effects at non-overlapping values due to a lack of data. To this end, the paper proposes an idea based on the prognostic score and identifiable representation (Intact-VAE). The prognostic score is an appropriate tool for limited overlapping because it can map some non-overlapping values to an overlapping value in a space of lower dimensions. Intact-VAE is a natural combination of iVAE and CVAE and is proposed to help to identify the causal effect. The paper implements this idea by proposing a new regularized VAE, called \beta-Intact-VAE, and further analyzes the treatment effect error bounds. In the end, the paper compares the model with recent methods in the experiments.

**Summary Of The Review:**

I vote for weak acceptance. The paper proposes a technically and theoretically sound approach and it studies an interesting setting -- limited overlapping, though it has some presentation issues.

---

> ### Author Response · Authors · 2021-11-18
> **Thank you for the review. We reply as follows.**
>
> (Presentation).
>
> Thanks again for the suggestions. We have **improved** the illustrations of concepts and assumptions. Please see the reply to all reviewers. The first 3 points in our reply to reviewer sh3V would also be interesting to you. On the symbols M (and P), they are special because they are the scores, and this is clearer in the new version; instead, we have **changed** the fonts for the real number and expectation. We will also add a figure illustrating the concept of overlap.
>
> (Experiments on PEHE).
>
> Measured by PEHE, our method is better than CFR on synthetic datasets with 1-dimensional true latent but is worse than CFR on IHDP. We believe our method can be better also on IHDP, if the representation is higher-dimensional/more flexible. The new experiments are running, and we will reply on this point as soon as we get the result.

---

> > ### Comment · Reviewer_ENDL · 2021-11-20
> > **Thanks for the reply**
> >
> > Thanks for the detailed replies to all reviews. I read through all comments and I believe one of the main concerns is about the paper presentation. I read briefly the new version and it looks better than the previous one. I have a question about the notation: $\mathbb{R}$ and $\mathbb{E}$ are typical in the papers and $\mathbb{P}$ and $\mathbb{M}$ are not often used for functions -- why not keep the typical ones? Did the notation follow previous papers?

---

> > > ### Author Response · Authors · 2021-11-22
> > > **Thank you for reading and affirming our replies and revision**
> > >
> > > (Symbols). We would like to have a font for the scores different from normal functions. However, we also understand your concern. Now, we use $\mathbb{R,E}$, but use $\mathfrak{p,m}$ (\mathfrak) for the scores.

---

> ### Author Response · Authors · 2021-11-22
> **Experiments for PEHE on IHDP**
>
> In additional experiments, we have found that our method works better on IHDP for PEHE, using higher-dimensional Z but smaller NNs to generate Z. Specifically, the results of (50 hidden units * 2 hidden layers) NNs in the prior and encoder, and with different Z dimensions are shown below:
>
> | $\dim(Z)$ | $\epsilon_{ate}$ | $\sqrt{\epsilon_{pehe}}$ |
> | --- | --- | --- |
> | 10 | 0.173 | 0.987 |
> | 20 | 0.166 | 0.844 |
> | 30 | 0.180 | 0.764 |
> | 40 | 0.186 | 0.716 |
> | 50 | 0.180 | **0.709** |
>
> Thus, we conclude that *our method performs better, or as well as, CFR (*$\epsilon_{ate}=0.25,\sqrt{\epsilon_{pehe}}=0.71$*) for both ATE and PEHE*. Moreover, it is promising that our method can get even better performance with more tuning of hyperparameters.
>
> We have **updated** Table 1 and its explanations. Please note that, the columns "Mod. *" have yet not changed to use the new parameters, and we will update as soon as we get the results.

---

### Official Review · Reviewer_ZUkW · 2021-11-02

**Correctness:** 4
**Technical Novelty And Significance:** 3
**Empirical Novelty And Significance:** 3
**Recommendation:** 6
**Confidence:** 3

**Main Review:**

Strengths:
- With the exception of a few errors, the paper is well written and has an understandable language. The motivation of the work is clear and relevant.
- The theoretical analysis in the paper is serious, as far as my understanding of the topic goes.
- The proposed method seems reasonable, and justified by the theoretical analysis in the paper.

Weaknesses:
- I found the paper to be a bit information dense. In fact, I am unsure about what could be some of the most subtle assumptions of the method. Clearly there is the assumption that X is a valid adjustment set, but what about (G1, additive noise models)? If this is necessary, then why use the machinery of a neural network? It seems that in order to make the theoretical analysis easier you have to assume a noiseless prior. Is there any chance you can make a list of the required assumptions, and a sentence or two of why they are needed? Maybe you can consider having such an explicit version of the assumptions in the main document or a separate section in the appendix.

- Along the same line of the previous point, I would invite the authors to think about scenarios in which such assumptions are fulfilled. The examples of this paper, in which we want to adjust a causal estimate using high dimensional covariates has always struck me as a bit unrealistic. As the authors might know, adjusting for all pre-treatment variables is not always the best option because that might introduce M-bias. I would like to say, though, that as the authors correctly point out, the more variables you have, the lower the chance of having overlap of the treatments. Can you please comment on what could be some possible scenarios in which the method could be applied?

- Finally, if known to the authors, I would like to know the relation of this work between work on generalization of neural networks. It seems, from my point of view, that the problem of limited overlapping of conditioning variables can be fixed by having estimators of the counterfactuals that generalize well beyond the support of the combination of covariates, and treatment. Could it be possible that what is driving most of your results is the regularization of the estimates? In general regularization is a way of achieving out of support generalization. Can you comment on this?


**Summary Of The Paper:**

The paper tackles the problem of estimating causal effects under partial overlap conditions. Overlap is a common assumption in causality, so the problem is hard and relevant. The authors present a theoretical analysis using PtS, an adaptation of prognostic scores. The estimation is carried out with a modified version of a Variational Autoencoder (VAE) called \beta-intact-VAE. In addition, the authors present bounds on the error of the Conditional Average Treatment Effect (CATE) using their framework. Their paper ends with experimental results of the proposed method and a comparison with other recent methods for CATE estimation.

**Summary Of The Review:**

I believe the paper solves the interesting problem of estimating CATE with limited overlap. It seems to me that most of the theory, and the estimation method were developed before this article, so in a certain way it is an incremental contribution. Nevertheless, both theoretical, and empirical analysis of the article seem serious.

---

> ### Author Response · Authors · 2021-11-18
> **Thank you for the affirmations and also the concerns. We reply to your questions/concerns below.**
>
> (Assumptions).
>
> To our understanding, even if under additive noise models, f* can be arbitrary complex and is better learned by NNs. Maybe your concern lies elsewhere. Could you explain?
>
> Indeed, the assumptions are scattered around the paper. Please see our reply to all reviewers for a commented list of assumptions.
>
> (Causal scenarios).
>
> To our understanding, here the reviewer raised two concerns about adjustment using high dimensional covariates: 1) there are subtle structures of the covariates (e.g., M-bias, see [1] for more), under which adjustment on all covariates introduces bias; and 2) "the more variables you have, the lower the chance of having overlap." In fact, our method addresses both issues.
>
> - First, on the structural issue, our (G1) rules out such bad controls, because, as specified by (G1), there are no unobserved variables that directly affect Y, except the exogenous noise e.
> - Second, our method recovers a low dimensional PtS M_t(X) as the representation Z, and this score/representation can satisfy overlap more easily due to the low dimensionality.
>
> We updated (G1) and the explanations below it, and we believe the causal scenario is much clearer. Please also see the point (Clarifications of assumptions) in our reply to all reviewers.
>
> [1] Cinelli, C., Forney, A., & Pearl, J. (2020). A crash course in good and bad controls. *Available at SSRN*, *3689437*.
>
> (Out-of-support generalization).
>
> To our understanding, the out-of-support generalization of NNs is not a fix to limited overlap. It is possible that NNs generalize to *some* non-overlapping values, but there is no guarantee to fix all non-overlapping values. On the other hand, our Sec. 3.2 provides the needed guarantee. On a related note, our method learns a low-dimensional representation, and the out-of-support generalization happens more likely in the representation space.
>
> (On contributions).
>
> We do not clearly understand the comment "most of the theory, and the estimation method was developed before this article" in the review summary.  We hope the reviewer explains it more with information on the previous literature, if this impacts the evaluation. Our first reply is as follows.  1) We believe our VAE architecture is novel and is far from "a modified version of" standard VAE. 2) As correctly pointed out by another reviewer, the hyperparameter \beta as a way to control balance is novel. 3) Our identification results in Sec. 3.2 is not found anywhere else, to the best of our knowledge. 4) The bounds in Sec. 4.2 is more related to previous works, but has novelties as mentioned in the 1st paragraph in Sec. 4.2.
>
> PS: Could you share the "a few errors" so that we can further improve the paper (although we believe they are minor or you would have made them explicit)?

---

> > ### Comment · Reviewer_ZUkW · 2021-11-25
> > **Clarification of some of the worries**
> >
> > I have read the newest version of the paper and I thank the authors for taking many of the comments from the reviewers into account. I can notice the improvement in the exposition of the paper. With respect to the author response:
> >
> > (Additivity)
> >
> > This comment was a result of a misunderstanding I had with respect to the relation between the prior and the noise from the additive model. For some reason, I understood that the prior of the VAE and e in (G1) were the same.
> >
> > (Causal scenarios)
> >
> > 1) I might not have made my question fully clear. What I meant to ask is whether the authors can think of a real-world scenario (no experiments needed) where the assumptions stated would hold well. The point of this question is that, from my perspective, people usually have problems finding variables to adjust their estimates instead of having a full set of variables that they already know
> > 2) Overlap: Yes, I think that the estimation under limited overlap is the biggest contribution of the paper, there was no question associated with this.
> >
> > (OOD generalization)
> >
> > Let me give you a trivial example where lack of overlap becomes a problem of OOD estimation. Suppose your data generation process is:
> > Y = aX + bT + cXT + e; Where the variable names follow roughly your notation. Now suppose that you observe X < 0 when T = 0, and X >= 0 when T = 1. If we are able to learn the parameters a, b, and c from the data available to us (this is possible using a simple linear regression), then we could estimate interventions and counterfactuals correctly. Now, I acknowledge this is a trivial example, but the point is that, in some cases, OOD generalization can fix the problem of non-overlapping variables.
> >
> > (On contributions)
> >
> > 1) On the identification: I agree that the identification results are novel, and the authors use previous results from the literature to build their proofs.
> > 2) On the estimation: Wu and Fukumizu (2021) developed intact-VAE, which is the backbone of the method proposed. As the reviewer pointed out, the \beta was proposed in the past (with disentanglement purposes). Yes, you use it for the purpose of controlling the balance, but no, I don't think this is a big architectural improvement over what is existent. This is, by no means, wrong or bad. I personally prefer simple methods that work elegantly for the problem and theoretical results at hand.
> > 3) On the bounds: I agree that these results are novel, but they are a result of the theory you developed before, so I can't see how anyone else could have come up with such bounds if you are the ones developing the theory associated to them.
> >
> > With respect to the smaller mistakes/typos:
> > - Page 4 "m(x) is a sufficient statistics" I believe it should be statistic instead.
> > I could not find more on this reread, but I will make sure to comment on them here if I find them.
> >
> > Thanks again to the authors for the interesting discussion.

---

> > > ### Author Response · Authors · 2021-11-27
> > > **Thank you so much for the clarifications and agreements. We further reply accordingly.**
> > >
> > > (Real-world scenario).
> > >
> > > We believe the real-world examples in Appendix C.2 are possible examples that would satisfy the assumptions. The analysis there focuses on the second part of (G2), but other assumptions are more easily satisfied. Let us consider them one by one (a kind reminder: a commented list of assumptions is in our "To all reviewers: Paper updated" ). (G1) additive noise model is common in real-world. (M1) is not an issue if injectivity holds in the true DGP. (D1) and (D2) are theoretical assumptions on the regularity of data and are very likely satisfied if X is randomly sampled. (M3') is not required in practice. Finally, (M2) is related to OOD generalization, please see the next point.
> > >
> > > We must admit that, without domain expertise, we are not sure about the above analysis. But we believe domain experts, e.g., the authors of the papers where the examples are taken from, can make quite definite judgments whether the assumptions are satisfied. In causal inference, domain expertise and careful prior investigations are always helpful.
> > >
> > > (OOD generalization)
> > >
> > > We agree this is a valid example. In fact, it is a special case of our (M2). In some sense, (M2) can be seen as an OOD generalization requirement: the NNs can learn the OOD score partition. This is indeed a very interesting connection. Our theory reduces a causal problem to an OOD generalization problem. Moreover, the requirement is weaker than common OOD generalization. This may open directions for future research. We will certainly add this discussion to the paper, possibly in the main text.
> > >
> > > (On contributions).
> > >
> > > On the architecture: We understand your concern. However, given that an arxiv paper is not regarded as a double submission by the policy of ICLR 2022, it would be uneasy if we are asked significant novelty from a recent arXiv paper.
> > >
> > > On the bounds: Yes, our bounds are based on our identifications. But, we were also inspired by the bounds in (Lu et al., 2020) (they are the "anyone else" who "come up with such bounds"). One of our contributions is to clarify and refine the bounds in (Lu et al., 2020) on more solid ground. Please see Appendix C.7 for detail.
> > >
> > > Thanks again for the active and interesting discussions. We will continue to fix the typos when spotted.

---

### Official Review · Reviewer_sh3V · 2021-11-02

**Correctness:** 3
**Technical Novelty And Significance:** 3
**Empirical Novelty And Significance:** 2
**Recommendation:** 6
**Confidence:** 3

**Main Review:**

## Strong points
- The authors extend the idea of prognostic scores to the context of machine learning and with the additional difficulty of limited overlap. Indeed this approach that can be seen as an alternative to the more common propensity score based approaches in causal inference is indeed interesting in cases of limited overlap which is relevant especially in high-dimensional settings.
- The authors provide a broad range of theoretical results, ranging from treatment effect identification via prognostic scores to a novel estimation strategy with error bounds for the CATE estimation.
- The article is well written, especially the first two sections lay out the context and motivation. The authors provide an adequately succinct yet sufficient list of references on related works to position their proposal.


## Issues/Points that require clarification
- As a general comment, the claims and statements of the main text, both in the identification and the estimation sections, are in part hard to follow without checking in the Appendix which makes the reading difficult. The content being very dense, I would suggest to either split the work into two articles, one for identification and the other on estimation or to submit this work under a more readable format or to defer the experiments to the appendix instead of the theoretical aspects and small examples which would help to follow the justifications in the identification and estimation sections.
- In definition 1 the authors talk about overlapping random variables ($V$) and in definition 2 they talk about functions. Do they mean functions of $x\in\mathcal{X}$ and when they say overlapping functions $\mathbb{P}_t(X)$, do they mean the random variables or do they make an implicit definition of overlapping functions?
- The authors claim that $\mathbb{M}(X)$ is an effect modifier. Since this relies on a definition of effect modifier from Hansen (2008) which is much less common than standard definitions of effect modifiers as interaction terms between covariates and the treatment assignment variable, it would be helpful to provide a concrete example (if not in the main text, then at least in the appendix) of such an effect modifier that does not involve $T$.
- At the end of Section 2, the authors provide an informal definition of _balanced PtS_. Why this non-rigorous definition instead of a formal one?
- Maybe I'm confusing the Intact VAE and the $\beta$ Intact VAE but from the graphical model of Intact VAE in Figure 1 I don't understand the connection between $Z$ and a PS $\mathbb{P}$. The definition of conditionally balanced representation requires that $Z\perp T|X$ which does not seem to be possible under the model of Intact VAE.
- The choice of the acronym "PS" for prognostic scores could be reconsidered since the target audience of this work seems to be rooted in causal inference where "PS" usually stands for the propensity score in many works.
- I have several questions about the experiments
    - Why the notation switch from $Z$ to $W$ in the experiments? Is there a conceptual difference between the two?
    - Why the choice of linear functions $h,k,l$ and how are they chosen?
    - Why isn't the proposed method from D'Amour and Franks (2021) also included in the comparative study? Even if their method relies on linear assumptions, it would be interesting to see a comparison in this context.
    - Since the paper contains a derivation of error bounds on the CATE (in Theorem 2), it would be interesting to assess them in the simulations and compare the empirical results to these bounds.
    - For the IHDP dataset, it is written that in the used model, $\beta=1$ (referenced as _Ours_ in Table 1) but then there are also modified versions that are reported with other values of the hyperparameter $\beta$. Shouldn't _Ours_ and _Mod.1_ show the same results or am I misunderstanding the table's description?


### Minor comments (that did not impact the score)
- p.1: Confounders are not necessarily high-dimensional. I would reformulate the last sentence of the 2nd paragraph to nuance the claim: rather say that the more covariates are collected the more likely unconfoundedness is to hold but that this can quickly lead to issues in separation between treatment and control groups.
- p.1: I would suggest adding the work on overlap weights by Li and Li (2019)
- p.3: In Section 2.1, I would recommend adding the SUTVA assumption for completeness (either by mentioning it directly or by adding the exclusion of interference in the assumptions).

######################################################################################################
### Post-rebuttal update
I thank the authors for their detailed and timely responses to all reviewers. Their efforts to clarify and reformulated assumptions on the DGP, model and data, as well as their concession to adapt their notations following the reviewers' remarks and the additional experiments related to the theoretical bound have helped addressing most of my concerns. However, I still think the positioning of their work relative to related works is difficult to follow or find which makes it difficult to compare its contributions with previous works. I have therefore decided to only increase my rating by one level.

######################################################################################################

### References
[1] Alexander D'Amour and Alexander Franks. Deconfounding Scores: Feature Representations for Causal Effect Estimation with Weak Overlap. _arXiv preprint arXiv:2104.05762_, 2021.

[2] Ben B. Hansen. The prognostic analogue of the propensity score. _Biometrika_ 95(2): 481-488, 2008.

[3] Fan Li and Fan Li. Propensity score weighting for causal inference with multiple treatments. _Annals of Applied Statistics_, 13(4):2389-2415, 2019.

[4] Christos Louizos, Uri Shalit, Joris Mooij, David Sontag, Richard Zemel, and Max Welling. Causal effect inference with deep latent-variable models. In _Proceedings of the 31st International Conference on Neural Information Processing Systems_, pp. 6449-6459. 2017.

[5] Uri Shalit, Fredrik D. Johansson, and David Sontag. Estimating individual treatment effect: generalization bounds and algorithms. In _International Conference on Machine Learning_, pp. 3076-3085. PMLR, 2017.

**Summary Of The Paper:**

The present paper proposes to address the issue of limited overlap in treatment effect estimation by investigating the identification assumptions of prognostic scores which, in certain contexts, are less restrictive on overlap than other methods such as propensity score based methods.
The main contributions of this work are presented in Sections 3.2 and 4.2, where they derive identifiability of the treatment effect via prognostic scores and propose a generative prognostic model based on variational auto-encoders (VAE).
The theoretical results are complemented with several synthetic and semi-synthetic experiments which also show that the proposed models can compete with and in certain cases improve upon state of the art.

**Summary Of The Review:**

In summary, the aim and approach of this paper are interesting and I believe the approach could be a methodological gain for the causal machine learning audience, but its underlying assumptions are presented in a way that makes it difficult to link or compare them to other works and its significance for real-world examples is therefore difficult to judge. Especially, the details of the underlying causal model are not clear to me and make it difficult to conceptually compare it with other methods such as CFR (Shalit et al., 2017) or CEVAE (Louizos et al., 2017).
I will read the rebuttal carefully and am willing to increase the score if the authors address the raised concerns.

---

> ### Author Response · Authors · 2021-11-18
> **Thank you so much for the thoughtful comments. Please have a look at the revision.**
>
> Thank you so much for the thoughtful comments. We are inspired to make some important clarifications related to the assumptions and causal model. Please see the reply to all reviewers for details. Also, we would love to hear what parts of the Appendix that you think should keep in the main text (Appendix C.3 contains several real-world examples, but they are rather detailed and hard to be kept in the main text). However, we are afraid that major structural revisions would change other reviewers' opinions (e.g., some reviewers think the experimental section is a strong point of this paper). We reply to all your specific points in another reply.

---

> > ### Comment · Reviewer_sh3V · 2021-11-19
> > **Short answer to respond to the authors' questions about restructuring and experiments**
> >
> > I thank the authors for the detailed response to my raised concerns and questions about the submitted work. Since the authors asked two concrete questions in their response, I now provide my answers to these questions before the end of rebuttal period to allow for a potential follow-up:
> >
> > - You are right that restructuring the manuscript given the restricted number of pages and the dense content of the current version is difficult. As a small change, I think that the example from Section C.1 could be easily moved to the main part and would have an impact since it provides a concrete small example for the motivation for the paper's focus on prognostic scores (instead of the more common balancing scores).
> > - Your suggestion to add results on the joint distribution of $(\mathbb{D}(x), \epsilon_f(x))$ seems pertinent and I would indeed be interested in examining these results.

---

> > > ### Author Response · Authors · 2021-11-22
> > > **Thank you for your kind understanding and suggestions**
> > >
> > > We have **updated** the paper to add the small example. Please see another reply on the experiments.

---

> ### Author Response · Authors · 2021-11-18
> **We reply to all your specific points below.**
>
>
> (Definition of Overlap).
>
> We only define overlapping RVs, and overlapping PtS means both the RVs P_0(X) and P_1(X) are overlapping. We have **updated** Def. 1 and the sentence below it. Now we avoid the word "function" and refer to the RVs directly.
>
> (M(X) in (G1)).
>
> We agree it is confusing to call M(X) effect modifier. On the one hand, it is in fact better to introduce interaction in M (**change** M(X) to M(X, T)). On the other hand, M(X, T) can be simply understood as a direct cause of y (**without** the need to mention "effect modifier"). We have **updated** the statement of (G1) and its explanations.   The term M_t(X) plays a key role in our clarifications on the causal model in the revised version. Please also see the point (Clarifications of assumptions) in our reply to all reviewers.
>
> (Balanced PtS).
>
> The naming "Balanced PtS" was not very good. Strictly speaking, a balanced PtS is just a PS, but low-dimensional PSs might not exist. Thus, for our purpose, here we would like to define "approximately balanced" PtS. The definition is informal because the notion "approximately balanced" is informal. We have **changed** the naming and **updated** the last paragraph of Sec. 2. With the updates, we believe that 1) it is clear that approximately balanced PtS means $\mathbb{P}_0 \approx \mathbb{P}_1$, and 2) the old "non-rigorous" notion becomes redundant, and it has been **removed**. Please also see the point (Score naming) in our reply to all reviewers.
>
> (Conditionally balanced representation and the generative model).
>
> This is indeed related to the difference between Intact-VAE and $\beta$-Intact-VAE. However, we would like to first note that conditionally balanced representation is possible under the model of Intact VAE. This requires the violation of *causal faithfulness*, i.e., there are other conditional independence relations (here Z⊥T|X), which are not generically implied by the graphical model. Our method, based on iVAE, which achieves ICA, performs nonlinear ICA to recover the scores. In fact, ICA procedures often violate causal faithfulness, because it requires finding causes from effects.
>
> Second, the violation of causal faithfulness is in fact not caused by the generative model (which is shown in Fig. 1), because the conditionally balanced representation is learned by the encoder and is enforced by $\beta$.
>
> Finally, there might be confusion between the DGP (G1) and the generative model of VAE. The former is the causal model, but the latter does not need to be causal. In our case, the generative model is built as a way to learn the scores through the correspondence to eq(2).
>
> We will add these explanations to Appendix C.2.
>
> (Acronym PS).
>
> We agree that PS would be confusing and have **changed** it. Please also see the point (Score naming) in our reply to all reviewers.
>
> (Conceptual comparisons to CFR and CEVAE).
>
> The comparisons with CFR were split around the paper, and the comparisons with CEVAE were mentioned in Sec. D.2. Now, we have **summarized and detailed** the comparisons in Sec. D.1.
>
>
> (Minor points).
>
> We have **added** the reference and **updated** the last sentence of the 2nd paragraph. We will also update on SUTVA, possibly in a footnote.
>
> (On Experiments below).
>
> (W and Z). Although Z models and recovers the true W, we intentionally distinguish them by the notation, because we often discuss both of them, e.g., dim(W) and dim(Z).
>
> (Linear functions). The linear functions have random coefficients. We use linear functions because then (M2) is satisfied straightforwardly. This choice is also not uncommon, e.g., in (Louizos et al., 2017).
>
> (D'Amour and Franks (2021)). It is not tried mainly because the code is not public. We have asked for the code by email on 10th Nov, but still have not received a reply. Also, the method estimates ATT in implementation and has hyperparameters, thus there would be difficult to adapt to our setting. We will ask again and add it to experiments if possible.
>
> (bounds). To our understanding, the reviewer wants to empirically check how tight are our bounds. This is indeed interesting, and we think it would be more relevant to focus on the $\mathbb{D}(x)$ term in Theorem 2 because it is directly related to conditional balance. We **plan** to plot the joint distribution of $(\mathbb{D}(x), \epsilon_f(x))$ (and can provide this result before the rebuttal deadline). Please kindly let us know your thoughts.
>
> (modified versions). "Mod. $\star$" does not mean "$\beta=\star$" (and "Ours" is not the same as "Mod. 1"). The modification is by adding unconditional balancing, and "$\star$" is the strength of unconditional balance (now have been **referred** as $\gamma$), while $\beta$ is the strength of our conditional balancing.

---

> ### Author Response · Authors · 2021-11-22
> **$(\mathbb{D}(x), \epsilon_f(x))$ plots**
>
> In the linked figure below, the rows correspond to 3 overlap levels from strong to weak ($\omega=6, 14, 22$ respectively). The first column shows the histograms of correlation coefficients (as "corrcoef" above each plot/histogram) on 100 random DGPs. The vertical bars in the histograms are 5, 25, 50, 75, 95 percentiles (the values are shown in the *table* below). The other 10 columns show the plots of the first 10 DGPs. We have two important observations from the histograms: 1) on the majority of DGPs, there are positive correlations between D and the error; 2) the positive correlation is stronger with weaker overlap (the portion of large correlation increases, and the mean corrcoefs are *0.100, 0.110, and 0.121*, respectively).
>
> Thus, our bounds and conditional balance have significance. Not all DGPs have positive correlations, and this is reasonable because our bound eq(11) has three other terms which can obscure the relation between D and the error. The DGPs *1, 3, 6, 8, 10* show typical distributions of $(\mathbb{D}(x), \epsilon_f(x))$ when there are positive correlations. The plots are in log-log scale, because both D and the error are single-sided, and most data points concentrate near $(0, 0)$, making the plots bad-looking.
>
> We will add a refined version of this analysis to the Appendix (and your thoughts are more than welcome).
>
> |  | 5 | 25 | 50 | 75 | 95 |
> | --- | --- | --- | --- | --- | --- |
> | $\omega=6$ | -0.289 | -0.086 | 0.069 | 0.299 | 0.609 |
> | $\omega=14$ | -0.328 | -0.124 | 0.055 | 0.337 | 0.636 |
> | $\omega=22$ | -0.274 | -0.128 | 0.067 | 0.341 | 0.634 |
>
> https://slack-files.com/TSCKL36SY-F02N6DZR6CT-8d088e852c

---

> ### Author Response · Authors · 2021-11-29
> **Thank you so much for raising the score!**
>
> And thanks for the affirmations of our replies and revisions!
>
> On related work, we will update the paper and add more pointers to the Appendix, and possibly move some important points to the main text.

---

### Official Review · Reviewer_3gmW · 2021-11-02

**Correctness:** 3
**Technical Novelty And Significance:** 3
**Empirical Novelty And Significance:** 2
**Recommendation:** 8
**Confidence:** 3

**Main Review:**

Strengths:

I believe the $\beta$-Intact-VAE is novel and is useful in treatment effects estimation.

The authors have provided theoretical guarantees for the proposed method.

Experimental results also demonstrate that $\beta$-Intact-VAE outperforms other generative models.


Weaknesses:

I am able to understand the general idea of the method but have difficulty understanding some notations in the paper. The authors might want to define some of the symbols before using them; for example,  $f_t$, $k_t$, $h_t$, $diag$ and $diag^{-1}$ in Section 3.1 and $f_t$, $k_t$, $h_t$ in Section 4.1. The notation $f_t$ indications that there are two different functions for $t=0$ and $t=1$, respectively, is it true?

I do not understand the differences between $p_\theta(y|x, t)$ and $p(y|x, t)$ in Section 3.2.

In the caption of Table 1, What is the unconditional balancing hyperparameter?

In Section 4.2, are the parameters $\Lambda$ learn from the data? In Equation (6), $ q_{\phi} (x, y, t) $ is still a function of $t$. How is balanced PtS achieved in this case? In the second term of Equatnion (7), it looks like $g_t^2(z)$ should be $g_t(z)$.

**Summary Of The Paper:**

This paper focuses on estimating treatment effects under a limited overlap, i.e., subjects with specific characteristics might only belong to one treatment group. To solve this problem, the authors propose a variational autoencoder (VAE) called $\beta$-Intact-VAE, which extends the framework of CVAE and iVAE.  The authors prove that the identifies the treatment effects. The proposed model is compared with other methods on synthetic datasets.


**Summary Of The Review:**

In general, I believe the $\beta$-Intact-VAE is novel and is useful in treatment effects estimation. The authors have provided theoretical guarantees for the proposed method. Experimental results also demonstrate that $\beta$-Intact-VAE outperforms other generative models.

---

> ### Author Response · Authors · 2021-11-18
> **Thank you for the favorable review. We make clarifications below.**
>
>
>
> Yes, f_t(x) is our shorthand for f(x, t); this is mentioned in footnote 2 (now in the paragraph below Def 2). We have **added/moved** the definitions of symbols as you suggested.
>
> The difference between pθ(y|x,t) and p(y|x,t) is important: the former is the model (as defined in eq(3)), and the latter is the truth (as defined in the 1st paragraph of Sec. 2.1).
>
> We modified our method by adding unconditional balancing, and the hyperparameter controls the strength. We have **updated** the paper to make this clearer.
>
> Yes, parameter $\Lambda$ is learned from the data. Please note that $\Lambda$ is just defined as the symbol for $\lambda$ when $\lambda_0 = \lambda_1$. We have **updated** the paper to make this clearer.
>
> The balanced PtS is achieved through \beta, which makes posterior more balanced, as detailed in the paragraph below eq(7). Indeed, if \beta is not very large, we will only have an "approximately balanced" PtS, and this is enough (and desirable, if there are no low-dimensional balanced PtSs in the DGP). On a related note, please see (Score naming) in our reply to all reviewers.
>
> The square on g_t in eq(7) was indeed a typo. We have **fixed** this and other related typos.

---

### Author Response · Authors · 2021-11-18
**To all reviewers: Paper updated**

We have updated the submission. The revision is not very large by word count, but we have put many thoughts into it (thanks to the inspiring reviews!) and believe it largely improves the clarity.

(Clarifications of assumptions).

We have **updated** the statements and explanations of (G1), (G2) (now (G1') ), and (G2') (now (G2) ). We also have **updated** the 1st paragraph of Sec. 4.1. Throughout, we stress the important role of function M (defined in (G1) ). This is now summarized by "the **essence of our method** is to recover the PGS M_t(X), assuming it is not higher-dimensional than Y and approximately balanced" at the end of Sec. 2.


(List of assumptions).

The following is a list of assumptions required by Theorem 1, with comments on their roles and subtleties. We hope this would aid a re-reading of the assumptions. Note that we also have **changed** two labels: (G2) → (G1') and (G2') → (G2). We use the new labels below.

- (G1) additive noise model. This is needed to ensure the existence of PtSs. (G1') is equivalent to (G1), and is introduced for better presentation, e.g., it connects to (G2) and (M1) through injectivity.
- (M1) and (D1) are inherited from iVAE and are required for model (parameter) identifiability (determines f_t up to *affine* mapping), which does not imply CATE identification in general. Arguably the most important is that the mapping f_t from latent Z to Y is *injective*, or else some information of Z is in principle unrecoverable.
- (M2), together with overlapping PtSs, is important to address limited overlap of X.
- (M3') means 1) we need to know or learn the distribution of hidden noise e and 2) noiseless prior. This simplifies the proof of identification, but when implementing the VAE as an estimation method, both noises are learned.
- (D2), or in fact (D2'), strengthens the model identifiability to "determines both f_0 and f_1 up to the *same* affine mapping", which replaces the balance of PS.

In addition,

- (G2) is required by Proposition 2 but not Theorem 1. It is no less important than (G1), because the *core intuition* of our method is that (G2) should hold approximately. Please see the updated 1st paragraph of Sec. 4.1. Appendix C.3 contains several detailed real-world examples on (G2).

We will add a refined version of this list to the Appendix.


(Score naming).

Following the comments of reviewer sh3V, we have re-considered the naming system and made the following **changes.**

1) PtS → PGS. There is no need to introduce a new term "PtS", although a caveat is that "PGS" often only refers to P_0(X), while we refer to P_0(X) and P_1(X) together.

2) PS → balanced PGS (bPGS). A PS is a conditionally balanced PtS.

3) balanced PtS → approximate bPGS. We used "balanced PtS" to also refer "approximately balanced PtS", and this was confusing. Now we distinguish them by "bPGS" and "approximate bPGS".

---

### Author Response · Authors · 2021-11-22
**To all reviewers: Second (minor) revision**

- In additional experiments, we have found that *our method performs better, or as well as, CFR for both ATE and PEHE on IHDP*, using higher-dimensional Z but smaller NNs to generate Z (previously worse than CFR for PEHE). We have updated Table 1 and its explanations. Please note that the columns "Mod. *" have yet not changed to use the new parameters, and we will update as soon as we get the results.
- Following the suggestion from Reviewer sh3V, the example from Section C.1 has been moved to the main text. It provides a better motivation for the focus on prognostic scores (instead of balancing scores).
- We also updated the code in case reviewers would like to check the new experiments in detail.

---

### Decision · Program_Chairs · 2022-01-20

**Decision:**

Accept (Poster)

**Comment:**

In this paper, the authors proposed a method for causal inference under limited overlap -- an important and understudied complication.  The authors propose to recover a prognostic score using a variational autoencoder, and thereby map a higher dimensional set of covariates with limited overlap to a lower dimensional set where overlap holds, and such that ignorability is maintained.

The paper was reviewed quite favorably by reviewers, and the authors updated the manuscript to address specific issues raised by reviewers.